# Ral GTPases promote breast cancer metastasis by controlling biogenesis and organ targeting of exosomes

Shima Ghoroghi[1,2,3], Benjamin Mary[1,2,3], Annabel Larnicol[1,2,3], Nandini Asokan[1,2,3], Annick Klein[1,2,3], Naël Osmani[1,2,3], Ignacio Busnelli[1,2,3], François Delalande[4], Nicodème Paul[2,3,5], Sébastien Halary[6], Frédéric Gros[1,2,3], Laetitia Fouillen[7], Anne-Marie Haeberle[8], Cathy Royer[9], Coralie Spiegelhalter[10], Gwennan André-Grégoire[11,12], Vincent Mittelheisser[1,2,3,13], Alexandre Detappe[13,14], Kendelle Murphy[15,16], Paul Timpson[15,16], Raphaël Carapito[2,3,5], Marcel Blot-Chabaud[17], Julie Gavard[11,12], Christine Carapito[4], Nicolas Vitale[8], Olivier Lefebvre[1,2,3], Jacky G Goetz[1,2,3†*], Vincent Hyenne[1,2,3,18†*]

[1]INSERM UMR_S1109, Tumor Biomechanics, Strasbourg, France; [2]Université de Strasbourg, Strasbourg, France; [3]Fédération de Médecine Translationnelle de Strasbourg (FMTS), Strasbourg, France; [4]Laboratoire de Spectrométrie de Masse BioOrganique (LSMBO), IPHC UMR 7178, CNRS, Université de Strasbourg, Strasbourg, France; [5]INSERM UMR_S1109, Genomax, Strasbourg, France; [6]CNRS, UMR 7245 MCAM, Muséum National d'Histoire Naturelle de Paris, Paris, France; [7]Université de Bordeaux, CNRS, Laboratoire de Biogenèse Membranaire, UMR 5200, Villenave d'Ornon, France; [8]Centre National de la Recherche Scientifique, Université de Strasbourg, Institut des Neurosciences Cellulaires et Intégratives, Strasbourg, France; [9]Plateforme Imagerie In Vitro, CNRS UPS 3156, Strasbourg, France; [10]IGBMC Imaging Center CNRS (UMR7104)/ INSERM (U1258)/ Université de Strasbourg, Illkirch, France; [11]Team SOAP, CRCINA, INSERM, CNRS, Université de Nantes, Université d'Angers, Nantes, France; [12]Integrated Center for Oncology, ICO, St-Herblain, France; [13]Nanotranslational laboratory, Institut de Cancérologie Strasbourg Europe, Strasbourg, France; [14]Équipe de synthèse pour l'analyse (SynPA), Institut Pluridisciplinaire Hubert Curien (IPHC), UMR7178, CNRS/Université de Strasbourg, Strasbourg, France; [15]Vincent's Clinical School, Faculty of Medicine, University of New South Wales, Sydney, Australia; [16]The Kinghorn Cancer Centre, Garvan Institute of Medical Research, Sydney, Australia; [17]C2VN, INSERM 1263, Inrae 1260, Aix-Marseille Université, Marseille, France; [18]CNRS SNC5055, Strasbourg, France

*For correspondence:
jacky.goetz@inserm.fr (JGG);
hyenne@unistra.fr (VH)

†These authors contributed equally to this work

Competing interests: The authors declare that no competing interests exist.

**Abstract** Cancer extracellular vesicles (EVs) shuttle at distance and fertilize pre-metastatic niches facilitating subsequent seeding by tumor cells. However, the link between EV secretion mechanisms and their capacity to form pre-metastatic niches remains obscure. Using mouse models, we show that GTPases of the Ral family control, through the phospholipase D1, multi-vesicular bodies homeostasis and tune the biogenesis and secretion of pro-metastatic EVs. Importantly, EVs from RalA or RalB depleted cells have limited organotropic capacities *in vivo* and are less efficient in promoting metastasis. RalA and RalB reduce the EV levels of the adhesion molecule MCAM/CD146, which favors EV-mediated metastasis by allowing EVs targeting to the lungs. Finally, RalA, RalB, and MCAM/CD146, are factors of poor prognosis in breast cancer

patients. Altogether, our study identifies RalGTPases as central molecules linking the mechanisms of EVs secretion and cargo loading to their capacity to disseminate and induce pre-metastatic niches in a CD146-dependent manner.

## Introduction

The communication between tumor cells and their neighboring stromal cells is essential to sustain tumor growth and promote invasion and metastasis (*Becker et al., 2016*; *Follain et al., 2020*). Notably, this communication allows tumors to indoctrinate their microenvironment and switch the phenotypes of various cell types, such as endothelial cells, fibroblasts, or immune cells to the benefit of tumor growth, invasion, immune escape and metastasis. Such communication occurs with organs distant of the primary tumors and favors the formation of pre-metastatic niches where the modified microenvironment can help settling metastatic tumor cells (*Peinado et al., 2017*). Seeding of this favorable metastatic environment can be mediated by soluble molecules (*Kaplan et al., 2005*; *Wang et al., 2017*) or by extracellular vesicles (EVs) secreted by tumor cells (*Costa-Silva et al., 2015*; *Hoshino et al., 2015*; *Jung et al., 2009*; *Peinado et al., 2012*). EVs are lipid bilayered vesicles of nanometric diameters containing a complex mixture of RNA and protein cargoes, including a repertoire of surface receptors (*Mathieu et al., 2019*). They can be directly secreted from the plasma membrane and called microvesicles or originate from an endosomal compartment, the multi-vesicular body (MVB), and then called exosomes (*van Niel et al., 2018*). The levels of circulating tumor EVs tend to correlate with tumor progression (*Baran et al., 2010*; *Galindo-Hernandez et al., 2013*; *Logozzi et al., 2009*). Accordingly, inhibition of key components of the EV secretion machinery often correlates with decreased metastasis (*Hyenne et al., 2017*). For instance, Rab27a, which directs exosome secretion by controlling the docking of MVBs to the plasma membrane (*Ostrowski et al., 2010*), promotes breast and melanoma tumor growth and metastasis in mice (*Bobrie et al., 2012*; *Peinado et al., 2012*) and predicts poor survival in human pancreatic cancer (*Wang et al., 2015*). In addition to the levels of secreted tumor EVs, their content, and in particular their set of surface adhesion proteins equally orchestrates metastasis formation. For instance, the presence of tetraspanins CD151 and Tspan8 on the surface of pancreatic adenocarcinoma EVs favors metastasis in rats by enhancing their adhesive capacities and controlling their biodistribution (*Yue et al., 2015*). Moreover, integrin receptors exposed by tumor EVs dictate their organotropism and thereby tune/control the seeding of a premetastatic niche in specific and distant organ (*Hoshino et al., 2015*). Therefore, accumulating evidence show that both the levels and the content of secreted tumor EVs are instrumental in promoting metastasis.

However, the molecular mechanisms coordinating these processes remain elusive. In particular, how the machinery governing EV secretion can impact the pro-metastatic properties of tumor EVs deserves in-depth characterization. To address this issue, we focused on the members of the Ral family, RalA and RalB (collectively referred to as RalA/B), acting downstream of RAS and promoting metastasis of different tumor types in both mice and human (*Gentry et al., 2014*; *Yan and Theodorescu, 2018*). We recently found that these versatile proteins are evolutionarily conserved regulators of exosome secretion (*Hyenne et al., 2015*). We originally observed that, in the nematode *C. elegans*, the Ral GTPase ortholog RAL-1 controls exosome secretion by acting on the biogenesis of MVBs. Importantly, we further showed that RalA/B modulate the levels of secreted EVs in models that are relevant to human breast cancer (*Hyenne et al., 2015*) suggesting that these GTPases could influence disease progression through EVs release. Here, we exploited 4T1 cells, an aggressive mammary tumor model that mimics human triple-negative breast cancer (*Kaur et al., 2012*) to further decipher how RalA/B tune EV secretion mechanisms and thereby control metastatic progression of the disease.

In this study, we first provide a detailed dissection of the impact of the Ral GTPases on EV secretion levels and unravel the mechanisms by which they control the homeostasis of MVBs. We have discovered that RalA/B directly acts through the phospholipase D1 (PLD1), which, as we show, also promotes EVs secretion, to favor the maturation of MVBs. We further demonstrate that RalA and RalB promote lung metastasis without affecting the invasive potential of breast carcinoma. Importantly, RalA/B are crucial for the organ targeting of tumor EVs, and, as a consequence, for the seeding of pre-metastatic niches. Finally, we identify the adhesion protein CD146/MCAM as a key EV

cargo controlled by RalA and RalB and demonstrate that it conveys, in part, the pro-metastatic function to EVs by controlling the lung tropism of breast cancer EVs.

## Results

### RalA and RalB control exosome secretion levels through the homeostasis of MVBs

We have previously shown that RalA and RalB control EV secretion in aggressive 4T1 mammary tumor cells (*Hyenne et al., 2015*) that reliably mimics the aggressive phenotype of human triple-negative breast cancer. We thus built on this relevant tumor model and decided to test the hypothesis that RalA and RalB could orchestrate pro-metastatic functions by tuning the molecular mechanisms driving the secretion levels and nature of EVs. We first confirmed our initial observations with the nanoparticle-tracking analysis (NTA) of EVs released by 4T1 cells and isolated by ultracentrifugation (100,000 g pellet). Stable depletion of RalA or RalB by shRNA reduces by 40% the amount of secreted EVs (*Figure 1a*, *Figure 1—figure supplement 1a*), with no impact on their average size (*Figure 1—figure supplement 1b*). RBC8 and BQU57, two previously described specific chemical inhibitors of Ral GTPases (*Yan et al., 2014*) significantly reduced EV secretion levels in mouse and human mammary tumor cell lines (4T1, MDA-MB231, D2A1, and MCF7 cells) as well as in two other cancer cell lines, human melanoma (A375) and pancreatic carcinoma (Panc1) cells (*Figure 1b* and *Figure 1—figure supplement 1c*). Together with evidence previously obtained in *Caenorhabditis elegans* (*Hyenne et al., 2015*), this demonstrates that the mechanisms by which RalA/B GTPases tune EV secretion levels are conserved throughout evolution and are notably at play in various cancer cell lines.

To better understand how Ral GTPases could impact EVs secretion, we first characterized their intracellular distribution in 4T1 cells. Endogenous RalA and RalB localize mostly within CD63-positive endosomal compartments (MVBs and late endosomes), as well as at the plasma membrane (*Figure 1c*). Similarly, GFP-tagged RalA and RalB localize both in late endosomal compartments positive for Lysotracker and at the plasma membrane (*Figure 1c*). Therefore, in 4T1 cells, Ral GTPases localize both at biogenesis sites of microvesicles (plasma membrane) and exosomes (MVBs). To further determine whether Ral GTPases affect MVBs as previously observed in *C. elegans*, we performed thorough electron microscopy (EM) analysis of endosomal compartments in 4T1 cells. In a first analysis of cells that were processed upon chemical fixation, we quantified the densities of (i) MVBs and (ii) endolysosomes, as well as (iii) the diameter of MVBs, (iv) the number and (v) the diameter of intraluminal vesicles (ILVs) per MVB. Strikingly, we found RalA or RalB depletion leads to a 40% decrease in the number of MVB per cytoplasmic surface in 4T1 cells (*Figure 1d* and *Figure 1—figure supplement 2a*), with no impact on the density of endolysosomes (*Figure 1—figure supplement 2b*). Further analysis of Lysotracker-positive compartments using FACS confirmed that RalA/B depletion has no significant effect on the late endosome-lysosome pathway (*Figure 1—figure supplement 2c*). Besides, EM analysis revealed no differences in ILV numbers per MVB surface (*Figure 1—figure supplement 2d*), nor in MVB diameters (*Figure 1—figure supplement 2e*). However, since chemical fixation is known to affect the morphology of endosomal compartments, we took our EM analysis one step forward by implementing high-pressure freezing (HPF) of cells, which better preserves the ultrastructure of endosomes (*Klumperman and Raposo, 2014*). A similar decrease in the number of MVBs per cytoplasmic surface in RalA and RalB knockdown cells was observed in these conditions (*Figure 1—figure supplement 2a*). Upon HPF, we further observed a slight decrease in the number of ILVs per MVB surface (*Figure 1—figure supplement 2d*) that could be, in part, explained by a slight increase in MVB diameters (*Figure 1—figure supplement 2e*). In conclusion, depletion of either RalA or RalB significantly reduces MVB number, while the remaining MVBs are slightly bigger. Overall, thorough EM analysis of intracellular compartments using both chemical fixation and HPF clearly demonstrates that both RalA and RalB control MVB homeostasis in breast mammary tumor cells.

### A RalA/B-PLD1-PA axis governs exosome biogenesis

We further investigated the molecular mechanisms controlling MVB homeostasis downstream of RalA/B GTPases. We decided to focus on phospholipases D (PLDs), which catalyzes the hydrolysis of

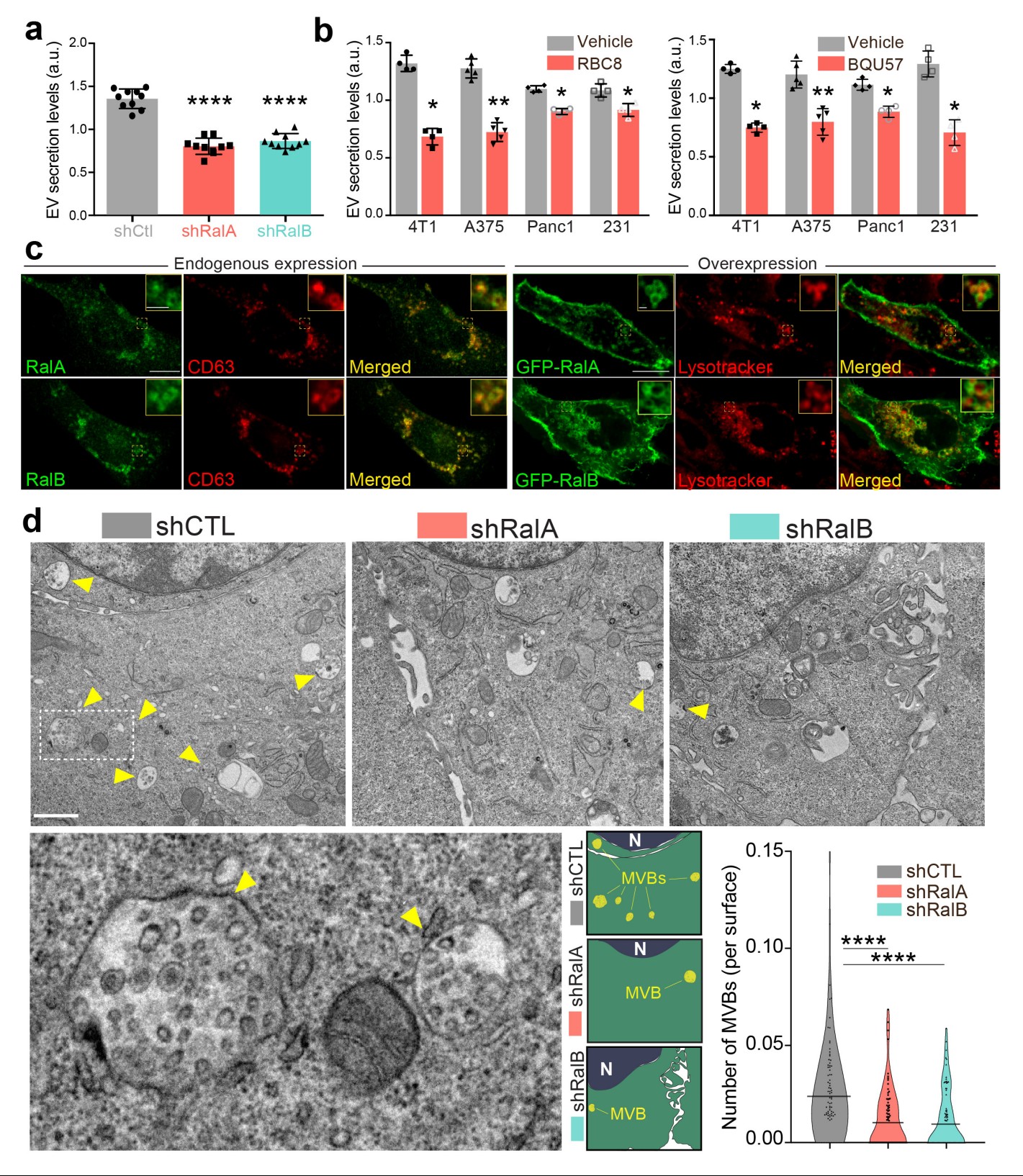

**Figure 1.** RalA and RalB control exosome secretion and multi-vesicular body (MVB) homeostasis. (**a–b**) Nanoparticle tracking analysis of extracellular vesicles (EVs) isolated by ultracentrifugation (100,000 g pellet) from the supernatant of shCtl, shRalA, or shRalB 4T1 cells (**a**) or from various cell types treated with Ral inhibitors RBC8 (**b**, left) or BQU57 (**b**, right). 231: MDA-MB-231 cells. Each dot represents one experiment (a: 10 independent

Figure 1 continued

experiments; One-Way Anova followed by Bonferroni's Multiple Comparison Test; b: four to five independent experiments, Mann Whitney test). (c) Representative confocal images of 4T1 cells showing endogenous expression of RalA, RalB, and CD63 by immunofluorescence (left) and overexpression of GFP-RalA and GFP-RalB in cells incubated with Lysotracker (right). Scale bar: 10 µm; zoom: 2 µm. (d) Representative electron micrographs of 4T1 shCtl, shRalA and shRalB cells, with zoom on MVBs; Scale bar: 1 µm; zoom: 200 nm. Violin plots show quantification of the number of MVB per cytoplasm surface. Each dot represents one field of view; horizontal bars represent the average (76–88 fields of view; Kruskal-Wallis test followed by Dunn's Multiple Comparison Test).

The online version of this article includes the following figure supplement(s) for figure 1:

**Figure supplement 1.** Ral knockdown efficiency and impact on EV secretion.
**Figure supplement 2.** Electron microscopy analysis of endosomes in the absence of RalA or RalB (a-b).

phosphatidylcholine (PC) into phosphatidic acid (PA), for three reasons: (1) PLD1 and PLD2 are two well-known targets of RalA and RalB (*Jiang et al., 1995*; *Luo et al., 1998*; *Vitale et al., 2005*), (2) PLD2 controls exosome secretion in breast cancer cells (*Ghossoub et al., 2014*), and (3) PLDs impact cancer progression (*Bruntz et al., 2014*). We first verified that both PLD1 and PLD2 are expressed in 4T1 cells by RT-qPCR (*Figure 2—figure supplement 1a*). In the absence of efficient anti-PLD antibody for immunofluorescence, we decided to assess the subcellular localization of PLD-GFP fusion proteins. PLD1 mostly localizes to endosomal compartments positive for RalA, RalB, and lysotracker, whereas PLD2 mostly localizes to the plasma membrane (*Figure 2a* and *Figure 2—figure supplement 1b*). Therefore, we tested whether PLDs could function downstream of RalA/B to control MVBs homeostasis and exosome secretion using two chemical inhibitors, CAY10593 for PLD1 and CAY10594 for PLD2 (*Lewis et al., 2009*; *Scott et al., 2009*). EM analysis of 4T1 cells revealed that inhibition of PLD1, but not of PLD2, induces a 40% decrease in the number of MVBs per cytoplasmic surface (*Figure 2b*). This phenotype is consistent with PLDs respective localizations and suggests that PLD1 functions in the RalA/B exosome secretion pathway. Further NTA analysis of treated cells showed that both inhibitors reduce EV secretion levels in 4T1 cells (*Figure 2c*), suggesting that both PLD isoforms regulate EV secretion potentially through distinct mechanisms. Importantly, PLD1 inhibition fully phenocopies the effect of RalA/B GTPases depletion, both on the cellular density of MVBs and on the level of EV secretion. To determine whether PLD1 acts downstream of RalA/B, we looked at its localization in the absence of RalA or RalB. Confocal analysis revealed that in 40% of shRalA or shRalB cells, PLD1 is uniformly cytoplasmic instead of being endosomal (*Figure 2d*). By contrast, RalA/B depletion had no major impact on PLD2 localization at the plasma membrane (also its trafficking might be altered) (*Figure 2—figure supplement 1c*). This shows that RalA/B GTPases are required for PLD1 localization on endosomes. To further investigate if PLD activity is involved in Ral GTPases-dependent EV secretion, we performed a lipidomic analysis of secreted EVs. As PLD converts PC into PA, we focused on these two lipid species. Importantly, RalA/B depletion significantly reduces the PA/PC ratio of secreted EVs (*Figure 2e*). In particular, the PA/PC ratio made of mono- and di-unsaturated lipid species (36:1, 36:2, 38:1, and 38:2), known to be PLD product/target, respectively, showed a tendency to be decreased although not reaching statistical significance (*Figure 2—figure supplement 1d*). This further implies that PLD's main product, PA, plays a crucial role in MVB homeostasis. Altogether, these results suggest that Ral GTPases control PLD1 localization on MVBs, which is required for local PA accumulation and ultimately for MVB homeostasis and exosome secretion (*Figure 2f*).

## RalA and RalB promote metastasis non-cell autonomously

Having identified RalA and RalB as important regulators of EV secretion in breast cancer cells, we next investigated whether such a function could impact metastasis. At first, we analyzed public databases to interrogate a potential correlation between RalA/B expression levels and metastatic progression. Using a large cohort of breast cancer patients with metastatic progression from the Cancer Genome Atlas (TCGA), we found that high expression of either RalA or RalB is significantly correlated with reduced survival (*Figure 3a*). Automated quantification of RalA/B expression levels by immunohistochemistry in primary tumors of breast cancer patients unraveled overexpression of both proteins in tumors from patients with metastasis (*Figure 3b*). These results prompted us to investigate in depth the role of RalA/B in a syngeneic mouse model of aggressive breast cancer, which is highly relevant to the human pathology.

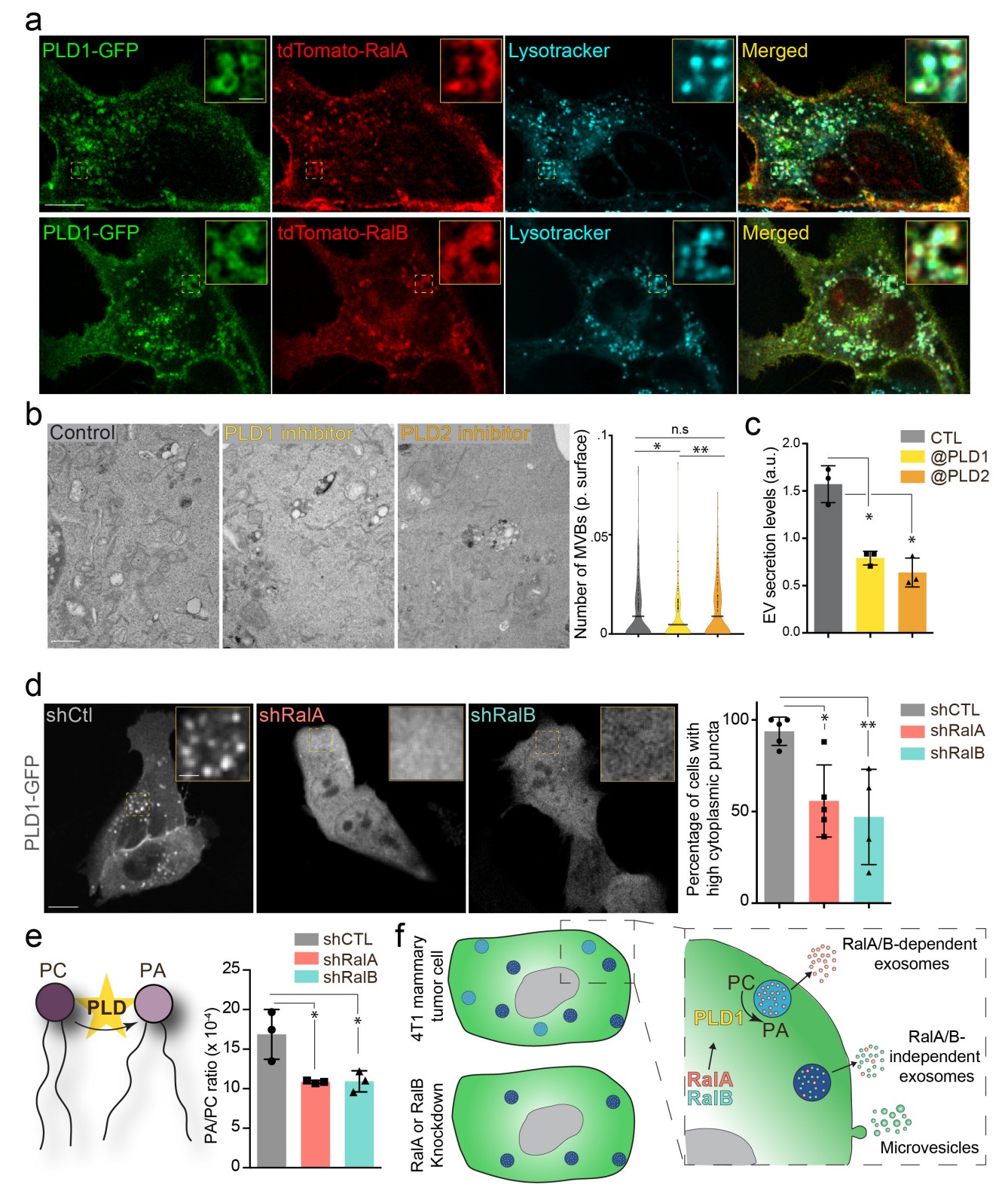

**Figure 2.** The RalA/B-PLD1-PA axis governs exosome secretion. (a) Representative confocal images of 4T1 cells co-transfected with PLD1-GFP and tdTomato-RalA (upper panels) or tdTomato-RalB (lower panels) and incubated with Lysotracker. Scale bar: 10 µm; zoom: 2 µm. (b) Electron microscopy analysis of 4T1 cells treated with PLD1 or PLD2 inhibitor. Scale bar: 1 µm. Violin plots show quantification of the number of multi-vesicular body (MVB) per cytoplasmic surface. Each dot represents one field of view; horizontal bar represents the average (180–194 fields of view; Kruskal-Wallis test

*Figure 2 continued on next page*

*Figure 2 continued*

followed by Dunn's Multiple Comparison Test). (**c**) Nanoparticle tracking analysis of extracellular vesicles (EVs) isolated by ultracentrifugation (100,000 g pellet) from the supernatant of 4T1 cells treated with PLD1 (CAY10593) or PLD2 (CAY10594) inhibitor. Each dot represents one experiment (three independent experiments; One-Way Anova permutation test followed by fdr multi-comparison permutation test). (**d**) Representative confocal images of shControl, shRalA and shRalB 4T1 cells transfected with PLD1-GFP. Scale bar: 10 µm; zoom: 2 µm. Graph shows the percentage of cells with high (>5) number of PLD1-GFP cytoplasmic puncta. (Each dot represents one experiment. Five independent experiments; Number of cells analyzed: shCtl (136), shRalA (170), shRalB (244); Kruskal-Wallis test followed by Dunn's Multiple Comparison Test). (**e**) Quantification of the Phosphatidic Acid (PA) / PhosphatidylCholine (PC) ratio in EVs isolated from shControl, shRalA, and shRalB cells (each dot represents one experiment; three independent experiments; One-Way Anova permutation test followed by fdr multi-comparison permutation test; fdr <0.1). (**f**) Model showing how RalA and RalB could control PLD1 localization on MVBs, thereby inducing the PA accumulation on MVBs, promoting MVB homeostasis and controlling exosome secretion.

The online version of this article includes the following figure supplement(s) for figure 2:

**Figure supplement 1.** PLD1 and PLD2 in 4T1 cells.

Therefore, we conducted a careful and exhaustive longitudinal analysis of metastatic progression of mammary tumors in syngeneic Balb/c mice. Briefly, 4T1 cells depleted or not for RalA or RalB were orthotopically grafted in mammary ducts, and several criteria were tracked over time. First, RalA and RalB have antagonist effects on tumor growth measured in vivo over time and ex vivo after 41 days: while RalA depletion significantly increased tumors growth, RalB depletion induced the opposite effect when compared to control tumors (*Figure 3c*). Neither RalA, nor RalB affected apoptosis, using caspase3 as a read-out (*Figure 3—figure supplement 1a–b*). In contrast, 4T1 cells depleted of RalA and RalB show increased growth rate *in vitro* and a decreased proportion of cells in sub-G1 phase of the cell cycle (*Figure 3—figure supplement 1c–d*). A similar increase in proliferation rates was observed *in vivo* in the absence of RalA (*Figure 3d*). Therefore, while depletion of RalA favors *in vivo* tumor growth by enhancing 4T1 proliferation potential, it is likely that additional non-cell autonomous factors are responsible for the decreased tumor growth observed upon RalB depletion.

We obtained the most striking result when carefully assessing the lung metastasis burden of these mice after 41 days. We measured the number and the surface covered by metastatic foci in serial lung sections and observed that RalA or RalB depletion in mammary tumors drastically reduced their metastatic potency (*Figure 3e*). When compared to the tumor growth rate, the most dramatic reduction of metastasis was observed in the case of RalA depletion. These experiments show that although RalA and RalB have antagonist effects on primary tumors, they both promote metastasis. To dissect this phenotype, we tested whether RalA or RalB could impact inherent cell migration and invasion potential of 4T1 cells, as it had been reported for RalB (*Oxford et al., 2005*; *Zago et al., 2018*). We performed 2D (*Figure 3f*) and 3D (*Figure 3g*) *in vitro* invasion assays and observed no effect of RalA or RalB expression levels on motility potential of 4T1 cells. Therefore, RalA/B seem to promote metastasis independently of cell invasion and are likely to promote metastasis of aggressive breast cancer cells non-cell autonomously by inducing pro-metastatic micro-environmental changes.

## RalA- and RalB-dependent EVs induce endothelial permeability

Since RalA and RalB promote metastasis independently of their cell-intrinsic properties, we wondered whether they could control secreted factors that are likely to induce micro-environmental alterations. In addition to EVs, tumor cells secreted soluble factors can promote metastasis by modulating the microenvironment, notably by promoting the formation of a metastatic niche (*Ombrato et al., 2019*). To test this possibility, we examined the impact of RalA and RalB on the soluble secretome of 4T1 cells. Depletion of RalA or RalB had no drastic effect on the soluble factors secreted by 4T1 cells (*Figure 3—figure supplement 2*). However, the secretion of one protein known to promote metastasis (*Ombrato et al., 2019*), WISP1/CCN4, is significantly decreased in shRalA/B cells (*Figure 3—figure supplement 2*). Thus, RalA and RalB are likely to enhance metastatic potency by promoting the secretion of EVs and possibly as well through WISP1/CCN4. Furthermore, in addition to enhancing the levels of secreted EVs, RalA/B could alter their functionality. To test this possibility, we challenged the pro-tumoral function of RalA/B EVs in an in vitro functional assay.

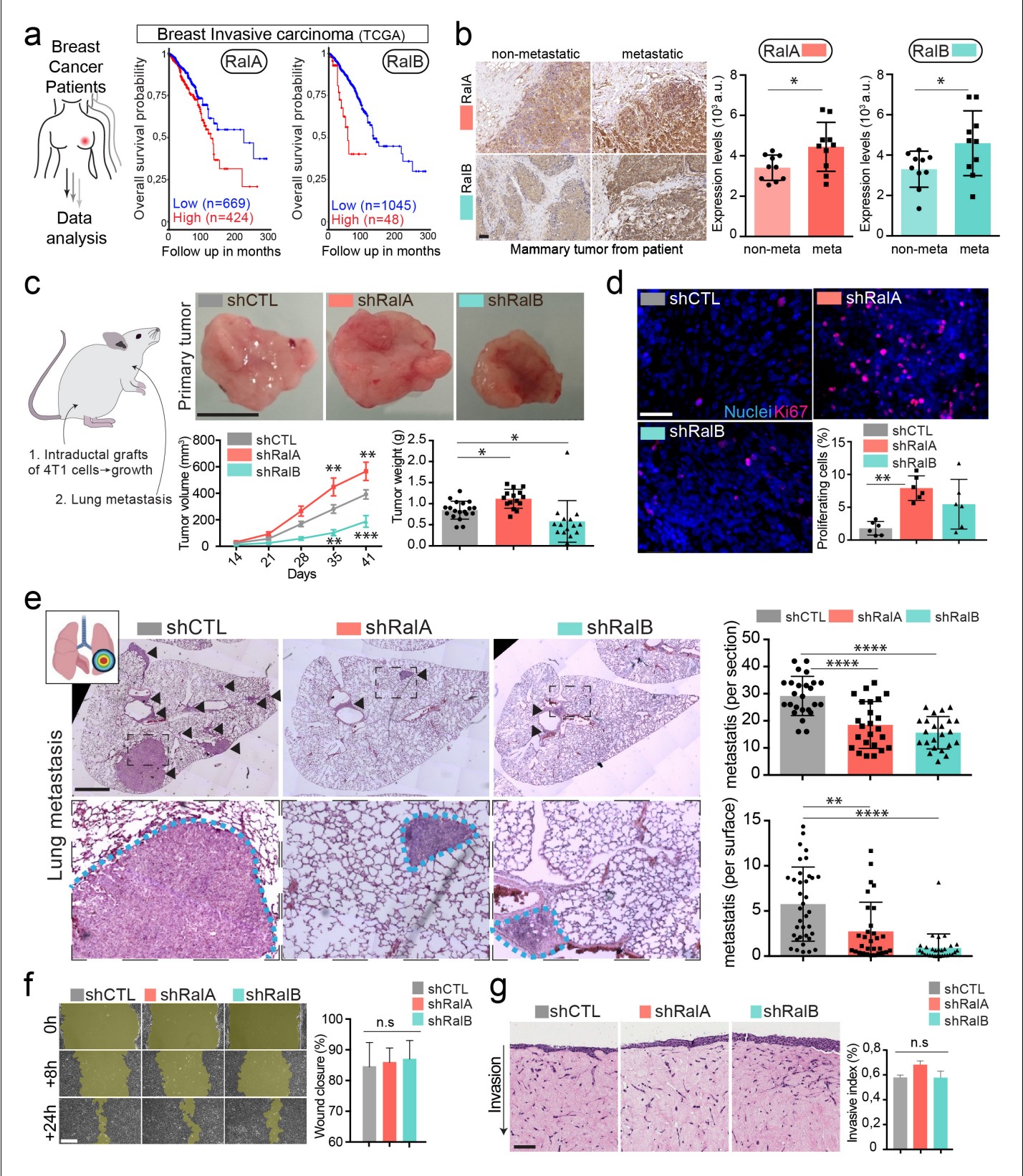

**Figure 3.** RalA and RalB promote lung metastasis in a non-cell autonomous fashion. (a) Kaplan-Meier curve, obtained from TCGA 1097 cohort, showing the survival probability of patients with tumor breast invasive carcinoma having high or low RalA (pvalue: 5.15 e-03; pAdj: 1.35e-01) or RalB (pvalue: 1.77 e-05; pAdj: 5.99e-03) expression levels. (b) Representative images of immunohistochemistry against RalA or RalB performed on mammary primary

*Figure 3 continued on next page*

**Figure 3 continued**

tumors from patients with or without metastasis. Scale bar: 500 µm Graphs represent automated scoring of DAB staining. Each dot represents one patient; 10 patients per group; Student t-test. (c) Orthotopic injection of shControl, shRalA, and shRalB 4T1 cells in syngenic mice. Representative images of primary tumors at day 41. Scale bar: 1 cm. Graphs showing the primary tumor growth over time (Left) and the primary tumor weight at day 41. Each dot represents one mouse. (Two independent experiments; Left: Two-way Anova followed by Bonferonni post-test, Right: Kruskal-Wallis test followed by Dunn's Multiple Comparison Test). (d) Representative images of primary tumors stained with anti-Ki67 antibody. Scale bar: 50 µm. Graph indicates the % of Ki67-positive nuclei. Each dot represents one mouse. (six mice taken from two independent experiments; Kruskal-Wallis test followed by Dunn's Multiple Comparison Test. (e) Analysis of lung metastasis in mice from the orthotopic experiment presented in (c). Representative images of lung sections (Day 41) stained with hematoxilin eosin. Scale bar: 1 mm. Graphs show the number of metastatic foci per section (upper, One-Way Anova followed by Bonferroni's Multiple Comparison Test) and the metastatic surface per lung surface (lower; Kruskal-Wallis test followed by Dunn's Multiple Comparison Test). Each dot represents one section f) Pictures of wound healing closure at different time points. Scale bar: 150 µm. Graph represents the percentage of wound closure at 16 hr (three independent experiments; Kruskal-wallis test followed by Dunn's multiple comparison test). (g) Pictures of 3D invasion assay after 15 days. Graph represents the invasive index. Scale bar: 100 µm.

The online version of this article includes the following figure supplement(s) for figure 3:

**Figure supplement 1.** Proliferation and apoptosis of 4T1 cells and tumors.

**Figure supplement 2.** Soluble secretome of 4T1 shControl cells compared to 4T1 shRalA or 4T1 shRalB cells (three independent experiments; One-Way Anova permutation test followed with pairwise permutation test with fdr correction).

Since tumor EVs are known to induce vascular permeability in the vicinity of tumors as well as in distant organs (*Tominaga et al., 2015*; *Treps et al., 2016*; *Zhou et al., 2014*), we tested the capacity of RalA/B dependent EVs to promote endothelial permeability *in vitro*. When added to a monolayer of endothelial cells, 4T1 EVs increased its permeability in a dose-dependent manner (*Figure 3—figure supplement 1e*). We then tested the impact of EV content on vascular permeability by subjecting endothelial cells to similar amounts of EVs derived from 4T1 cells expressing or not RalA/B. Interestingly, endothelial monolayers became less permeable when treated with a similar amount of EVs derived from shRalA or shRalB cells. Similarly, such EVs fail to disrupt adherent and tight junctions by contrast to EVs derived from 4T1 control cells (*Figure 4b*) suggesting that EVs from RalA/B knockdown cells have reduced pro-permeability abilities. Therefore, depletion of RalA/B reduces secretion levels of EVs and leads to the secretion of EVs whose effect on vascular leakiness is hampered. The important observation that vascular permeability could be reduced upon depletion of RalA or RalB, and with a similar amount of EVs, prompted us to further dissect whether RalA or RalB could tune the priming of pre-metastatic niches.

## RalA and RalB dependent EVs are pro-metastatic and lung tropic

Here, we thus explored whether RalA and RalB synergistically impact the pro-metastatic functions of EVs by tuning their secretion levels as well as their content. Since on one hand RalA and RalB positively control the levels and the functionality of secreted tumor EVs (*Figures 1* and *4a*), and on the other hand they promote metastasis (*Figure 3*), we tested a direct impact of RalA/B-dependent EVs on the promotion of lung metastasis. For this, we decided to directly assess the role of 4T1 EVs in priming lung metastatic niches in vivo, as previously described for other tumor EVs (*Costa-Silva et al., 2015*; *Hoshino et al., 2015*; *Peinado et al., 2012*; *Zhou et al., 2014*). Priming of lungs with control EVs significantly enhances lung metastasis over 14 days when compared to PBS (*Figure 4c*). In striking contrast, priming of mouse lungs with a similar number of EVs derived from Ral-depleted cells did not promote metastasis. This key experiment demonstrates that RalA/B confer pro-metastatic functions to EVs, in addition to controlling their secretion levels. Indeed, the decreased metastasis observed in absence of RalA/B can result from either drastically reduced EVs secretion or diminished pro-metastatic potential of EVs. To unravel why EVs from RalA/B-depleted cells are unable to promote metastasis, we first determined their capacity to efficiently reach the lungs and prime pre-metastatic niches by tracking the dissemination of fluorescently labeled EVs that were injected in the blood circulation of Balb/c mice. We found that 1 hr after injection 4T1 EVs mostly accumulate in the lungs, as well as the liver and brain (*Figure 4d* and *Figure 4—figure supplement 1a*). These three organs are the main metastatic organs of 4T1 cells, and breast carcinoma, showing that the organotropism of 4T1 EVs mirrors the metastatic organotropism of their parental cells and further validates the relevance of our model to human pathology (*Kaur et al., 2012*; *Lou et al., 2008*). Through a careful analysis of cell types that internalize EVs in these conditions, we

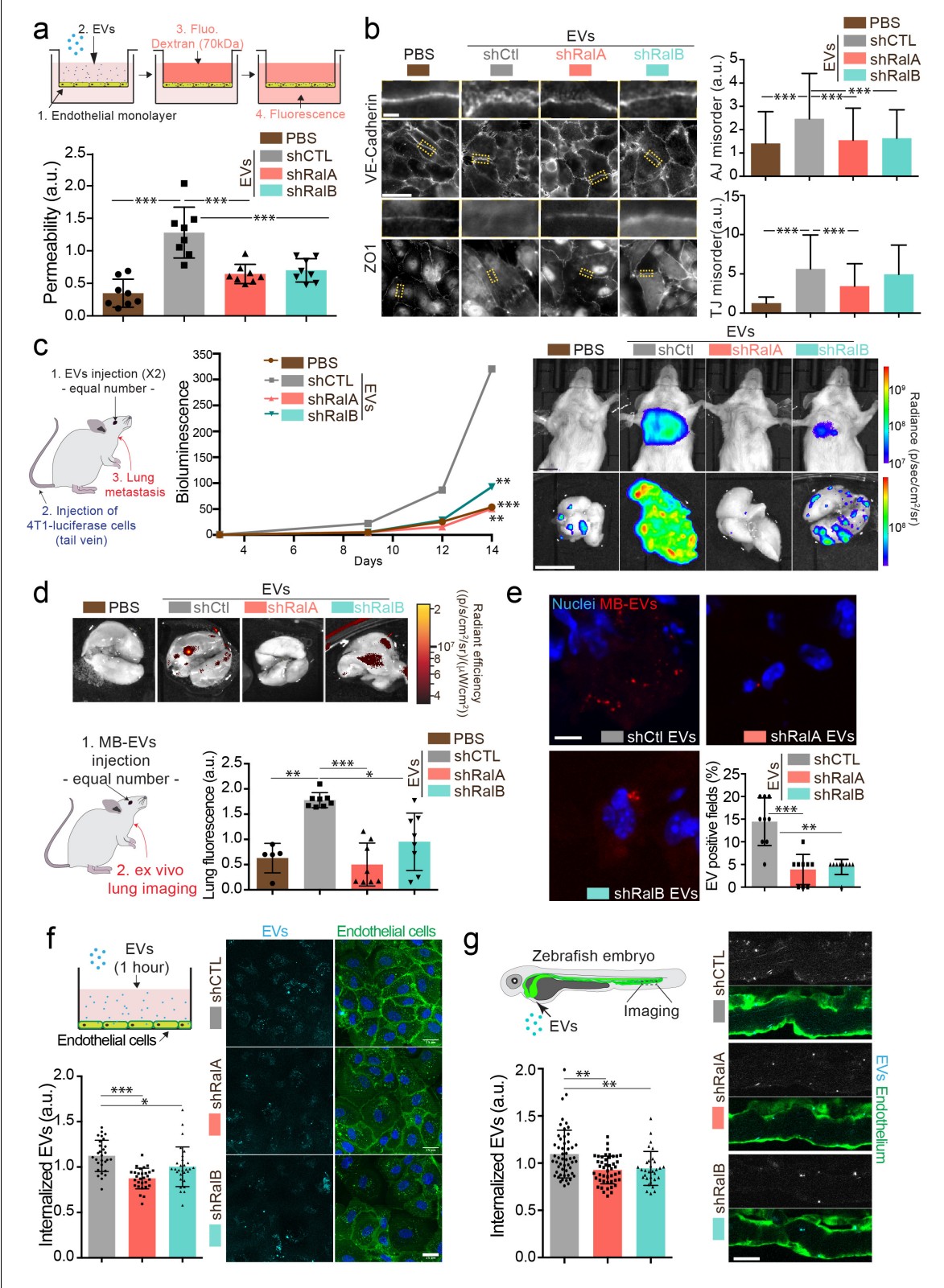

**Figure 4.** RalA and RalB control lung tropism of pro-metastatic tumor extracellular vesicles (EVs). (a) Effect of a similar amount of EVs on HUVEC monolayer permeability *in vitro*. The graph represents the normalized amount of fluorescent dextran that crossed the endothelial barrier. Each dot represents one experiment (eight independent experiments; One-Way Anova followed by Bonferroni's Multiple Comparison Test). (b) Representative epifluorescence images of VE-cadherin (upper panels) and ZO1 (Lower panel) stainings on HUVECS cells treated with similar amounts of EVs. Scale bar:

*Figure 4 continued on next page*

*Figure 4 continued*

20 µm; zoom: 2 µm. Graphs represent the disorganization of adherent (up) and tight (low) junctions (Three independent experiments; up; Kruskal-Wallis test followed by Dunn's Multiple Comparison Test). (c) Metastasis priming experiment, Balb/c mice are first injected twice with tumor equal number of EVs ($1.5 \times 10^8$ EVs), then intravenously with 4T1 luciferase cells and metastasis is then followed over time. Graph shows metastasis progression over time in mice pre-injected with PBS, or with equal number of EVs from shControl, shRalA or shRalB cells (7–10 mice per group; merge of two independent experiments; Two-way Anova followed by Bonferonni multiple comparison post test; stars indicate statistically significant differences at day 14). Right: *In vivo* and *ex vivo* representative images of mice and lungs at day 14. Scale bars: 1 cm. (d–e) Lung accumulation of equal number of fluorescent-labeled EVs ($3 \cdot 10^8$ EVs), from shControl, shRalA or shRalB cells injected intravenously. (d) Representative ex vivo images and graph showing the total lung fluorescence 1 hr post-injection. Each dot represents one mouse. (Eight mice taken from two independent experiments; Kruskal-Wallis test followed by Dunn's Multiple Comparison Test.) (e) Representative confocal lung sections images and graph showing the percentage of EVs-positive fields. Each dot represents one section (three mice; Kruskal-Wallis test followed by Dunn's Multiple Comparison Test). Scale bar: 5 µm. (f–g) Arrest and internalization of equal number of EVs from shControl, shRalA, and shRalB cells on endothelial cells *in vitro* and *in vivo*. (f) Representative confocal Z-stacks of equal number of EVs after 1 hr or incubation with HUVEC monolayer. Scale bar: 25 µm. Each dot represents one field of view (each dot represents one field of view from three independent experiments; Kruskal-Wallis test followed by Dunn's Multiple Comparison Test). (g) Representative confocal Z-stacks the caudal plexus of Tg(Fli1:GFP) zebrafish embryos, where GFP is expressed in the endothelium, injected with similar number of EVs and imaged right after injection. Each dot represents one zebrafish (31–53 embryos from four independent experiments; Kruskal-Wallis test followed by Dunn's Multiple Comparison Test). Scale bar: 20 µm.

The online version of this article includes the following figure supplement(s) for figure 4:

**Figure supplement 1.** 4T1 extracellular vesicles (EVs) organotropism.

observed that 4T1 EVs mostly accumulate in endothelial cells, macrophages, and fibroblasts of the lung parenchyma (*Figure 4—figure supplement 1b*). Importantly, EVs derived from RalA- or RalB-depleted cells failed to efficiently reach the lungs, even though similar amounts were injected in all conditions (*Figure 4d,e*). Similar results were observed for EVs reaching the liver (*Figure 4—figure supplement 1c*). Hence, we can conclude at this stage that RalA/B control the pro-metastatic properties of EVs by tuning their ability to reach vascular regions and local parenchyma and efficiently reach metastatic organs, thereby modulating the formation of a pre-metastatic niche.

The latter results raised the exciting hypothesis that metastasis impairment could be, in part, explained by a general defect in adhesion of circulating EVs at the vascular wall. We recently showed that EVs target specific vascular regions by first arresting at the surface of endothelial cells (*Hyenne et al., 2019*). We used two complementary models that allow careful tracking of single EVs and assessed early events of EVs internalization in endothelial cells. Using microfluidics, we found that internalization of 4T1 EVs within endothelial cells is decreased after 1 hr when they originate from RalA/B-depleted cells (*Figure 4f*). Similarly, upon tracking of fluorescent EVs injected in the circulation of zebrafish embryos, we observed that endothelial arrest/internalization of EVs from RalA/B knockdown cells is significantly hampered (*Figure 4g*). Altogether, these experiments suggest that RalA/B knockdown significantly reduced the adhesive properties of EVs to the endothelium, establishing a potential link with their failure to accumulate in mice lungs. Furthermore, our results support a model in which RalA/B GTPases, in addition to promoting EV secretion, also control the pro-metastatic function of these EVs, likely by modulating their content.

## RalA/B promote CD146 EV loading for efficient lung targeting and pre-metastatic niche priming

These functional experiments (*Figure 4*) suggest that the content of EVs can directly influence metastasis formation and that such content is likely to be impacted by RalA/B. Therefore, we carried out a careful and thorough molecular comparison of the cargo content of EVs derived from RalA/B-tuned cells. We first analyzed the RNA content of EVs using RNAseq and found that a large proportion of the RNAs present in EVs from shRal cells were different from the control (30–50%) (*Figure 5a*; *Supplementary file 1*). Accordingly, GO terms associated with mRNA enriched in each EV type showed important differences in biological processes, molecular function, or cellular components (*Figure 5—figure supplement 1*). In addition, EVs from shRalA cells differed from control or shRalB EVs in the nature of the RNA they contain, as shRalA EVs showed an important increase in non-coding RNA (*Figure 5b*). Overall, this experiment reveals that RalA/B have a profound impact on the content of RNA in 4T1 EVs.

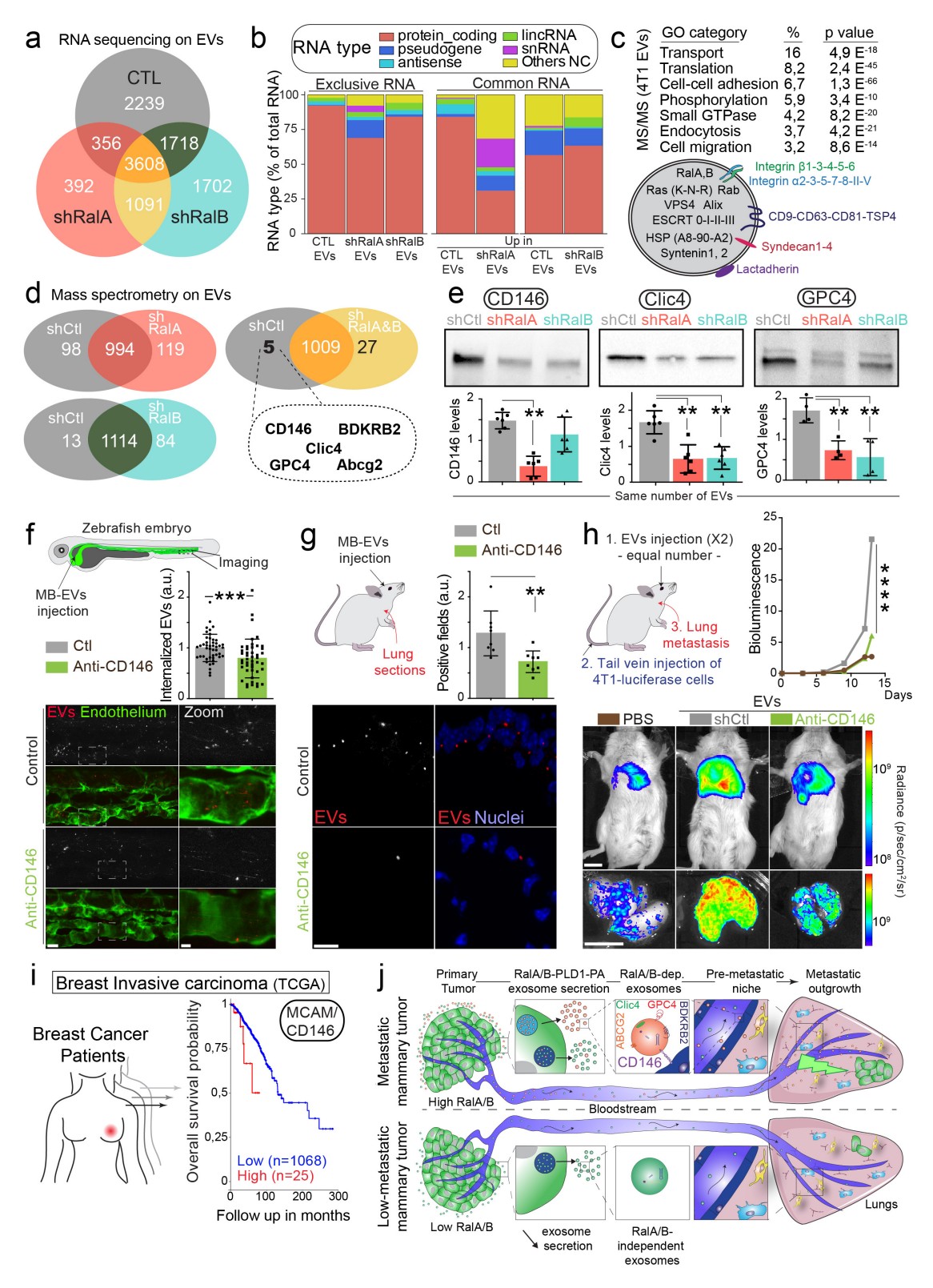

**Figure 5.** CD146/MCAM is under-expressed in RalA/B knockdown extracellular vesicles (EVs) and mediates their lung tropism. (**a**) Venn diagram representing the RNA present in the EVs isolated from shControl, shRalA or shRalB cells (with a minimum of 10 reads per sample; RNA sequencing performed in triplicate). (**b**) Type of RNA associated identified in EVs isolated from shControl, shRalA, or shRalB cells. Left: RNA exclusively present in one type of EVs. Right: enriched RNAs (log2 fold change >2; p(adj.)<0,05). (**c**) GO terms of the proteins identified in EVs isolated from 4T1 cells by

*Figure 5 continued on next page*

Figure 5 continued

ultracentrifugation (100,000 g pellet) and illustration of some proteins known to be present in EVs. (d) Comparison of the protein content of EVs isolated from shControl, shRalA, and shRalB cells. The venn diagram represents proteins having different expression levels (Mass spectrometry performed in triplicate; FDR < 1%). (e) Analysis of the expression of CD146/MCAM, Clic4, and Glypican4 in EVs isolated from shControl, shRalA and shRalB cells by western blots. Each dot represents one experiment (four to six independent experiments; Kruskal-Wallis test followed by Dunn's Multiple Comparison Test). (f–g) Arrest, internalization, and organotropism of EVs treated with an anti-CD146 antibody and injected in the circulation of zebrafish embryos (f) or mouse (g). (f) Representative confocal Z-stacks the caudal plexus of Tg(Fli1:GFP) zebrafish embryos, where GFP is expressed in the endothelium, injected with equal number of EVs and imaged right after injection. Scale bar: 20 µm; Zoom scale bar: 5 µm. Each dot represents one zebrafish (46 embryos from four independent experiments; Mann Whitney test). (g) Representative confocal images of lung sections and graph showing the percentage of EVs-positive fields. Scale bar: 10 µm. Each dot represents one mouse (eight mice from two independent experiments; Mann Whitney test). (h) Metastasis priming experiment, Balb/c mice are injected twice with tumor equal number of EVs ($1.5 \times 10^8$ EVs), pre-incubated with CD146 blocking antibody or isotype control, and then intravenously injected with 4T1 luciferase cells and metastasis is followed over time. Graph shows metastasis progression over time (14 mice per group; merge of two independent experiments; Two-Way Anova followed by Bonferonni multiple comparison post-test; stars indicate statistically significant differences at day 14). *In vivo* and *ex vivo* representative images of mice and lungs at day 14. Scale bars: 1 cm. (i) Kaplan-Meier curve, obtained from TCGA 1097 cohort, showing the survival probability of patients with tumor breast invasive carcinoma having high or low MCAM/CD146 expression levels (pvalue: 3.42 e-02; pAdj: 5.67e-01). (j) Model describing the role of RalA/B-dependent EVs in metastatic formation.

The online version of this article includes the following figure supplement(s) for figure 5:

**Figure supplement 1.** RNA content of extracellular vesicles (EVs) from shControl, shRalA, and shRalB cells.

**Figure supplement 2.** 4T1 cells and extracellular vesicles (EVs) express CD146/MCAM long isoform.

We further analyzed the protein content of 4T1 EVs by mass spectrometry. As shown in *Figure 5c* and 4T1 EVs contain a large number of proteins usually found in small EVs (77 of the top 100 proteins from Exocarta are found in 4T1 EVs; *Supplementary file 2*), such as tetraspanins, integrins, ESCRT proteins or small GTPases, such as RalA/B themselves. Importantly, many of these proteins are known to localize to endosomes, suggesting that some of these EVs are bona fide exosomes. Unexpectedly, comparison of the proteome of EVs secreted by RalA or RalB knockdown cells did not reveal major differences, as no protein is exclusive to one type of EVs. Instead, a small proportion of proteins showed differential expression levels (*Figure 5d*; *Supplementary file 2*). Regarding their protein content, we noted that EVs from control cells are closer to EVs from shRalB cells (97 proteins with differential expression) than to EVs from shRalA cells (217 proteins with differential expression). We then focused on the five proteins over-expressed in EVs from shCtl cells compared to both EVs from shRalA and EVs from shRalB cells. These proteins are CD146/MCAM, Clic4, Glypican 4, BDKRB2, and Abcg2. We verified the expression levels of CD146/MCAM, Clic4 and Glypican4 by western blot of identical number of EVs (*Figure 5e*). While Clic4 and Glypican4 are significantly under-expressed in EVs from shRalA or shRalB cells, the long isoform of CD146/MCAM (*Figure 5— figure supplement 2a*) showed a significant decrease in EVs from shRalA cells, and a tendency to decrease in EVs from shRalB cells, which was confirmed by anti-CD146 ELISA (*Figure 5—figure supplement 2b*). The hypothesis that Ral GTPases could control CD146 EV loading is further sustained by colocalization analysis. Indeed, by immunofluorescence, we observed that CD146 localizes both at the plasma membrane and in CD63-positive MVB/late endosomes in 4T1 cells, similarly to Ral GTPases (*Figure 5—figure supplement 2c*). Altogether, content analysis reveals that depletion of either RalA or RalB deeply affects the EV RNA loading and changes the levels of several key proteins.

We next interrogated whether the impact of RalA/B on the lung targeting and priming potential of EVs could be explained by its effect on the EV levels of MCAM/CD146. MCAM/CD146 (also known as Mel-CAM, Muc18, S-endo1, Gicerin) is an adhesion receptor overexpressed in various cancer types, including breast cancer, where it was shown to promote invasion and tumor progression (*Garcia et al., 2007*; *Zeng et al., 2011*; *Zeng et al., 2012*). In addition, MCAM/CD146 is present on endothelial cells where it mediates the adhesion of several cell types, including the transendothelial migration of monocytes (*Bardin et al., 2009*). Given, the known function of MCAM/CD146 in cell adhesion (*Wang and Yan, 2013*), we hypothesized that it may, at least in part, be responsible for the lung tropism defects observed with EVs derived from RalA/B-depleted cells. To test the involvement of MCAM/CD146 in EVs adhesion, we treated 4T1 EVs with an anti-mouse MCAM/CD146 blocking antibody before injection in zebrafish or mouse circulation. EVs pretreated with MCAM/

CD146 blocking antibody failed to successfully arrest on endothelial walls of zebrafish embryos (*Figure 5f*) and inefficiently reached the lungs in our mouse model (*Figure 5g*). Finally, we assessed the functional role of EV-bound CD146 in priming of pre-metastatic niches. To do this, 4T1 EVs were pre-treated with MCAM/CD146 blocking antibody (or with an isotype control) and injected intravenously, preceding tail-vein injection of 4T1 luciferase cells. Blocking CD146 on EVs significantly reduced their pro-metastatic potential. Therefore, inhibition of MCAM/CD146 precludes their lung accumulation and the subsequent formation of metastasis and thereby phenocopies RalA/B knockdown. These results demonstrate that MCAM/CD146, whose presence at the surface of EVs is tuned by RalA/B, is, at least partly responsible of the adhesion and lung tropism of 4T1 EVs. It further explains why EVs from RalA knockdown cells, which have reduced levels of MCAM/CD146, fail to reach the lungs efficiently. The pro-metastatic role of MCAM/CD146 is further confirmed by the analysis of a human cohort of breast cancer showing that its high expression is associated with worsened prognosis (*Figure 5h*). Altogether, our work demonstrates that RalA/B, by controlling MVB homeostasis, promote the secretion CD146-enriched EVs, whose lung tropism sustains efficient metastasis (*Figure 5i*).

## Discussion

The therapeutic limitations of breast cancer metastasis warrant a deeper understanding of its molecular machinery. Our findings highlight the exosome-mediated priming of metastatic niches by Ral GTPases as a critical requisite for lung metastasis during breast cancer progression. We show that RalA and RalB promote the secretion of exosomes by maintaining a high number of multi-vesicular bodies, likely through the PLD1-PA axis. Furthermore, we demonstrate that RalGTPases favor the secretion of CD146-rich exosomes, which accumulate in metastatic organs, notably in lungs, where they establish premetastatic niches (*Figure 5i*). Finally, we show that high levels of RalA and RalB correlated with poor prognosis suggesting a unified mechanism for human breast cancer metastasis.

This work, together with our previous study of RAL-1 in *C. elegans* (*Hyenne et al., 2015*), establishes Ral GTPases as major evolutionarily conserved mediators of exosome secretion. Our experiments suggest that RalA/B contribute to exosome secretion in several tumor cell lines, of different origins, implying that they might function pleiotropically over various cancers. Our results suggest that RalA/B and their effector PLD1 affect the levels of secreted exosomes by tuning the levels of cytoplasmic MVBs. While Ral GTPases, partially localized at the plasma membrane, could also affect microvesicle secretion, our data indicate that they function in exosome biogenesis upstream of PLD1. Similarly, a direct correlation between MVB density and levels of secreted EVs was recently suggested by studies showing that chemical or electric stimulation of MVB biogenesis results in increased EV secretion (*Kanemoto et al., 2016*; *Yang et al., 2020*). The formation of MVBs results from dramatic biochemical transformations of endosomes involving multiple protein and lipid switches (*Huotari and Helenius, 2011*; *Scott et al., 2014*). Understanding the steps at which RalA/B and PLD affect this endosome maturation program is critical and remains to be fully deciphered. Our results from mice and *C. elegans* suggest that biogenesis of ILVs, which is a key step in MVB maturation and the initial phase of the exosome secretion pathway, could as well be controlled by RalA/B. Our work further identifies PLD as an effector acting downstream of Ral to control exosome secretion. Whether other Ral effectors contribute to EV secretion remains to be addressed. Interestingly, while PLD2 was found to impact exosome secretion by governing ILV biogenesis in a different breast carcinoma cell line (*Ghossoub et al., 2014*), our data rather suggest that PLD1 controls exosome biogenesis in 4T1 cells. Indeed, PLD1 localizes on MVBs and its inhibition, but not the inhibition of PLD2, decreases MVB density. Nevertheless, it should be noted that we measured EV secretion levels and MVB density based on PLD inhibition at previously-published high concentrations of the inhibitors (compared to their respective IC50) and that off-target effect can not be ruled out. By contrast, PLD2 is essentially localized at the plasma membrane of 4T1 cells and its inhibition reduces EV secretion suggesting that PLD2 could rather promote microvesicle secretion in 4T1 cells. Therefore, we speculate that RalA/B-PLD1 control ILV biogenesis in 4T1 cells, possibly through the regulation of PA levels. Alternatively, they could impact the homeostasis of a subclass of MVBs, for instance by controlling their stability or their degradation.

Priming of metastatic niches by (soluble or) EV-mediated factors takes central stages in cancer progression (*Gao et al., 2019*; *Peinado et al., 2017*) and identification of molecular machineries

that underlie this condition could point to new therapeutic or diagnostic targets. Our study demonstrates that Ral GTPases enhance the formation of lung metastasis in mouse models, by promoting the secretion of exosomes within primary tumors, while RalA/B expression levels correlates with metastasis in human breast cancer. While the pro-tumoral activity of Ral GTPases was so far mostly attributed to their capacity to promote anchorage-independent cell growth (for RalA) or cell invasion (for RalB) (*Yan and Theodorescu, 2018*), we now show that Ral GTPases have additional non-cell autonomous functions, and that these functions are important contributors to metastasis. Indeed, in 4T1 cells, depletion of either RalA or RalB alters the levels, content and functionality of secreted EVs, without decreasing cell migration or proliferation. Depending on the cell type or the biological process, RalA and RalB can display redundant, synergistic or even antagonist activities (*Gentry et al., 2014*). Since RalA and RalB mostly share similar phenotypes regarding EV secretion, content and function, they likely function in the same pathway. Interestingly, both Ral proteins appear to be essential for exosome secretion, revealing that their functions are not fully redundant. Therefore, both GTPases are required for the generation of a specific subpopulation of EVs with enhanced pro-metastatic properties and further work is needed to fully unravel the downstream molecular pathways. With this work, RalA and RalB add to the list of proteins known to control exosome secretion and to affect tumor progression, such as Rab27a (*Bobrie et al., 2012*; *Kren et al., 2020*; *Peinado et al., 2012*), Alix (*Monypenny et al., 2018*), syntenin (*Das et al., 2019*), and components of the ESCRT machinery (*Mattissek and Teis, 2014*). These studies demonstrate that the number of EVs secreted by a primary tumor is an essential element determining the efficiency of metastasis. However, it is important to keep in mind that all these proteins regulating EV trafficking, including RalA/B, contribute to tumor progression through both exosome dependent and exosome independent functions. Altogether, despite pointing to additional functions of RAL GTPases, our study is the first to identify new molecular machinery from its function in EV biogenesis up to its pro-metastatic function in breast cancer lung metastasis.

Priming of metastatic niches by EVs has, so far, mostly been attributed to increased levels of pro-metastatic EVs with pro-metastatic functions (*Becker et al., 2016*; *Bobrie et al., 2012*; *Peinado et al., 2012*). In addition to controlling the levels of secreted EVs, we show that RalA/B affect their function by enhancing their capacity to induce endothelial permeability *in vitro* and pre-metastatic niches *in vivo*. These two observations could be linked, as RalA/B-dependent EVs could promote endothelial permeability locally in the primary tumor or at distance in lungs, thereby favoring both tumor intravasation and extravasation. Content analysis revealed that RalA/B control the identity and levels of RNAs and proteins present in secreted EVs. Interestingly, Ras, which is known to activate RalA/B (*Gentry et al., 2014*), also controls the protein and RNA cargo of tumor EVs (*Cha et al., 2015*; *Demory Beckler et al., 2013*; *McKenzie et al., 2016*), although its effect on the levels of secreted EVs is unclear (*Demory Beckler et al., 2013*; *McKenzie et al., 2016*). As McKenzie and collaborators identified a MEK-ERK-Ago2 pathway downstream of Ras (*McKenzie et al., 2016*), it would be interesting to determine how this pathway connects with the Ral-PLD-PA axis described in our study. Among the few proteins significantly enriched in RalA/B-dependent EVs, we identified CD146, a molecule known to modulate cell-cell adhesion (*Wang and Yan, 2013*). We showed, using functional inhibition, that CD146 present on pro-metastatic EVs controls their lung targeting efficiency thereby impacting their biodistribution and niche-promoting function. Accordingly, we and others show that high expression of CD146 correlates with poor prognosis in human breast carcinoma (*Garcia et al., 2007*; *Zeng et al., 2012*). CD146 functions as an adhesion molecule involved in homophilic and heterophilic interactions (*Wang and Yan, 2013*), promoting for instance monocyte transmigration (*Bardin et al., 2009*). CD146 can perform trans-homophilic interactions via its immunoglobulin-like extracellular domain (*Taira et al., 1994*; *Taira et al., 2005*). It also binds to extracellular matrix proteins or other transmembrane proteins, such as VEGFR2 (*Wang and Yan, 2013*). Therefore, it is tempting to speculate that CD146 affects the biodistribution and organ targeting efficiency of circulating tumor EVs by mediating their interaction with specific ligands present on the luminal side of endothelial cells of metastatic organs. Other adhesion molecules, such as integrins and tetraspanins were shown to affect the biodistribution of tumor EVs and ultimately the formation of metastasis (*Hoshino et al., 2015*; *Yue et al., 2015*). Therefore, it is likely that the combination of these receptors at the surface of tumor EVs, combined with the differential expression of their ligands on endothelial cells throughout the organism will dictate their homing. More work will be needed to characterize this organ specific EV zip code and to identify relevant endothelial ligands

for circulating EVs and develop inhibitory strategies to impair their arrest and uptake at metastatic sites. In addition, the presence of other cell types in the circulation, such as patrolling monocytes, which take up large amounts of circulating EVs, could also contribute to the accumulation of tumor EVs in specific organs (*Hyenne et al., 2019*; *Plebanek et al., 2017*). Finally, other factors, such as the vascular architecture and hemodynamic patterns could be involved (*Follain et al., 2020*; *Hyenne et al., 2019*) and the interplay between these mechanical cues and the surface repertoire of metastatic EVs should be a fertile ground for future research. Precisely dissecting the mechanisms by which tumor EVs reach specific organs would allow to understand the priming of premetastatic niches.

Overall, our study identifies RalA/B GTPases as a novel molecular machinery that regulates the formation and shedding of pro-metastatic EVs. We also discovered CD146 as an EV cargo whose targeting could inspire new therapeutic strategies to impact the progression of metastatic breast cancer.

## Materials and methods

### Cell culture

The establishment of 4T1 cell lines stably expressing shRNA against RalA, RalB, or a scramble sequence has been described previously (*Hyenne et al., 2015*). 4T1-Luciferase (RedLuc) cells were purchased from Perkin-Elmer. All 4T1 cell lines were cultured in RPMI-1640 medium, completed with 10% fetal bovine serum (FBS, Hyclone) and 1% penicillin-streptomycin (PS) (GIBCO). 4T1 shRNA cell lines were maintained in medium containing 1 µg/ml puromycin, except during experiments, and regularly checked for the stability of knockdown by western blots. Human Umbilical Vein Endothelial Cells (HUVEC) (PromoCell) were grown in ECGM (PromoCell) supplemented with a supplemental mix (PromoCell C-39215) and 1% PS. Human A375 melanoma and human MDA-MB-231, MCF7 and SKBR3 breast cancer (ATCC) cell lines were grown in high-glucose Dulbecco's modified Eagle's Medium (DMEM, Gibco Invitrogen Corporation) supplemented with 10% (FBS) and 1% PS. D2A1 cell were grown in high-glucose Dulbecco's modified Eagle's Medium (DMEM, Gibco Invitrogen Corporation) supplemented with 5% (FBS), 5% new born calf serum, 1% non-essential amino acids, and 1% PS. Human Panc-1 pancreatic adenocarcinoma cell line was grown in RPMI-1640 supplemented with 10% FBS, and 50 µg/ml gentamicin sulfate (Gibco/Life Technologies). All cell lines were cultured in a humidified atmosphere containing 5% $CO_2$ at 37°C and checked regularly for absence of mycoplasma by PCR (VenorGeM, Clinisciences).

### Plasmid transfections

Cells at 50–70% confluency were transfected with 1 µg of plasmid using JetPRIME (PolyPlus, Illkirch, France) according to the manufacturer's instructions. The following plasmids were used: pGFP-PLD1, pGFP-PLD2 (*Corrotte et al., 2006*), pLenti CMV:tdtomato-RalA, and pLenti CMV:tdtomato-RalB.

### Drug treatment

Cells were incubated with the following drugs in the appropriate medium: RalA/B inhibitors BQU57 (10 µM; Sigma) and RBC8 (10 µM; Sigma), PLD1 inhibitor CAY10593 (10 µM; Santa Cruz Biotechnology) or PLD2 inhibitor CAY10594 (10 µM; Santa Cruz Biotechnology). Cells were treated for 18 hr before processing for EV isolation or cell analysis.

### qRT-PCR analysis

Total RNA was extracted from cells using TRI Reagent (Molecular Research Center) according to the manufacturer's instructions. For qRT-PCR, RNA was treated with DNase I and reverse transcribed using the High-Capacity cDNA RT Kit. qRT-PCR was performed using the Power SYBR Green PCR Master Mix or TaqMan Gene Expression Master Mix using a 7500 Real-Time PCR machine (Applied Biosystems). All compounds were purchased from Life Technologies (St Aubin, France). Data were normalized using a Taqman mouse probe against GADPH as endogenous control (4333764T, Life Technology) and fold induction was calculated using the comparative Ct method (-ddCt).

## Western blot

Cell or EV extracts were denatured in Laemmli buffer and incubated at 95℃ for 10 min. 10 μg of protein extract (for cell lysates) or equal number of EVs ($8.50 \times 10^8$ EVs per lane, measured by NTA) were loaded on 4–20% polyacrylamide gels (Bio-Rad Laboratories, Inc). The following antibodies were used: CD9 (Rat, 553758; BD Biosciences), RalA (mouse, 610221; BD Biosciences), RalB (mouse, 04037; Millipore), Glypican 4 (Rabbit, PA5-97801; Thermo Fisher Scientific), antibodies specifically recognizing the short and long isoforms of CD146 were previously described (*Kebir et al., 2010*), Clic4 (mouse, 135739; Santa Cruz Biotechnology), α-tubulin (mouse, CP06; Millipore) and Secondary horseradish peroxidase-linked antibodies: anti-Rat (GE healthcare; NA935), anti-Mouse (GE healthcare; NA 931) and anti-rabbit (GE healthcare; NA934). Acquisitions were performed using a PXi system (Syngene). Intensities were measured using the Fiji software.

## Elisa

Elisa was performed according to the manufacture's instruction (RayBiotech) by loading equal number of EVs ($7 \times 10^8$ - $9.5 \times 10^9$) per well (two experiments in triplicate).

## Electron microscopy

### Chemical fixation

Cells were fixed with 2.5% glutaraldehyde/2.0% paraformaldehyde (PFA) (Electron Microscopy Sciences) in 0.1M Cacodylate buffer at room temperature for 2 hr, then rinsed in 0.1M Cacodylate buffer (Electron Microscopy Sciences) and post-fixed with 1% OsO4 (Electron Microscopy Sciences) and 0.8% $K_3Fe(CN)_6$ (Sigma-Aldrich) for 1 hr at 4℃. Then, samples were rinsed in 0.1M Cacodylate buffer followed by a water rinse and stained with 1% uranyl acetate, overnight at 4℃. The samples were stepwise dehydrated in Ethanol (50%, 70% 2 × 10 min, 95% 2 × 15 min and 100% 3 × 15 min), infiltrated in a graded series of Epon (Ethanol100%/Epon 3/1, 1/1, 1 hr) and kept in Ethanol100%/ Epon 1/3 overnight. The following day, samples were placed in pure Epon and polymerized at 60℃. One hundred nm thin sections were collected in 200 copper mesh grids and imaged with a Philips CM12 transmission electron microscope operated at 80 kV and equipped with an Orius 1000 CCD camera (Gatan).

### High-pressure freezing

HPF was performed using an HPF COMPACT 03 high pressure freezing machine (Wohlwend), using 3 mm diameter Aclar film disks (199 um thickness), as cell carriers. Subsequent freeze substitution in acetone was performed using an automatic FS unit (Leica AFS), including 0.25% OsO4 staining, and Epon embedding. Sections were contrasted on grids with 1% uranyl acetate followed with 0,4% lead citrate (Sigma-Aldrich). Imaging was performed similarly to chemical fixation.

The number of MVBs and lysosomes per surface of cytoplasm were quantified using the Fiji software. MVBs and lysosomes were distinguished based on their morphology: MVBs have one or more ILVs and lysosomes contain ILVs but are also electron dense and contain irregular membrane curls.

## FACS analysis

For cell cycle analysis, $10^6$ cells were fixed using the FoxP3 Staining Kit (00-5523-00 eBioscience) for 30 min at room temperature in the dark. Samples were then resuspended in permeabilization buffer containing 20 μg of RNase A (R6513 Sigma) and 1 μg of propidium iodide (PI) (130-093-233 Miltenyi Biotech) for 30 min. PI fluorescence was analyzed using a BD Accuri C6 cell analyzer with BD CSampler Analysis Software. Results were analyzed with FlowJo software version 10 (TreeStar).

For lysosomal analysis, confluent cells were incubated with 1 μM Lysotracker Green DND 26 (L7526-Thermo Fischer) diluted in complete RPMI medium for 30 min at 37℃. Cells were then detached by addition of TrypLE (12604021, ThermoFischer), washed in PBS 2% (v/v) FCS, and stained with 0.1 μM DAPI in PBS 2% (v/v) FCS immediately before analysis. Samples were processed on a Gallios Flow Cytometer (Beckman Coulter). Dead cells and doublets were excluded from analysis respectively by the selection of DAPI negative cells and co-analysis of integral vs time-of-flight side scatter signals. Data were analyzed on FlowJo software (BD Bioscience). Mean Fluorescence intensities (MFI) of lysotracker in each condition were normalized by performing a ratio with MFI of an unstained condition in the same channel.

## Migration assays

For 2D migration assays, 4T1 mammary tumor cells were plated on 35 mm plastic dishes (six-well plates) and grown for 2 days until reaching 90% confluence. The cells were then grown for 16 hr in serum-free medium before wounding of the monolayer by scraping from the middle of the plate. Cells were incubated in complete RPMI medium and sequential images of the wound were collected with a ×10 objective at 0, 8, and 24 hr after wounding. Percentage of wound closure over time was analyzed and quantified using the Fiji software.

3D Organotypic invasion assays were conducted as previously described (*Timpson et al., 2011*; *Vennin et al., 2017*). Briefly, rat tail tendon collagen was extracted with 0.5 mol/L acetic acid to a concentration of 2.5 mg/ml. A total of $8.4 \times 10^4$ telomerase immortalized fibroblasts (TIFs) were embedded into the neutralized collagen in the presence of 1 x MEM and 8.8% FBS. Matrices were allowed to contract over a 12-day period in DMEM (1% P/S, 10% FBS). Following contraction TIFs were removed with puromycin (2 μg/ml) for 72 hr before $8 \times 10^4$ 4T1 cells were seeded on the contracted matrices and allowed to grow to confluence for 48 hr in RPMI (1% P/S, 10% FBS). The matrices were then transferred to an air-liquid interface on a metal grid and cells allowed to invade for 15 days with media changes every 2 days. Following the invasion, organotypic matrices were fixed in 10% buffered formalin and processed for histochemical analysis. The invasive index was measured in three representative fields of view per matrix with three matrices per replicate for three replicates.

$$\text{Invasive Index} = \frac{\text{Number of cells} > 200\ \mu m\ \text{depth}}{\text{Cells on top of the matrix}}$$

## *In vitro* permeability assay

Transwell filter inserts (pore size 1.0 μm, 12 mm diameter, polyester membrane, Corning, New York, USA) were coated with fibronectin (10 μg/ml; Sigma). Then, HUVECs were seeded ($0.3 \times 10^6$ cells/well) and grown on transwell filters for 48 hr until reaching confluency. Confluent monolayers of HUVEC cells were treated with similar amounts (10–100 μg) of 4T1-EVs, PBS (as a negative control) or with 100 ng/ml TNF-α (as a positive control) overnight. FITC-dextran (MW ~70,000; Sigma) was added to the top well at 25 mg/ml for 20 min at 37°C, and fluorescence was measured in the bottom well using a fluorescence plate reader (Berthold Tris Star 2; 485 nm excitation and 520 nm emission). Cells were washed for three times and were fixed for immunofluorescence (described below).

## Secretome analysis

Cell culture supernatants were collected and centrifuged for 15 min at 300 g. Supernatants were incubated with Mouse XL Cytokine Array membranes (R and D Systems) according to the manufacturers' instructions. Three independent experiments were performed. Intensities were measured using the Fiji software.

## *In vitro* proliferation assay

Briefly, cells were seeded in 96-well plates at the density of 2000 cells per well with 200 μl of complete culture medium and cultured for 24, 48, and 72 hr at 37°C. Culture medium without cells was used as the blank control group. To avoid the edge effect, the peripheral wells were filled with sterile PBS. For the proliferation test, a total of 20 μl MTS solution was added to each well, followed by incubation for 2 hr at 37°C. Optical density was measured at 490 nm using a Berthold Tristar device.

## EVs isolation and characterization

Cells were cultured in EV-depleted medium (obtained by overnight ultracentrifugation at 100,000 g, using a Beckman, XL-70 centrifuge with a 70Ti rotor) for 24 hr before supernatant collection. The extracellular medium was concentrated using a Centricon Plus-70 centrifugal filter (10 k; Millipore) and EVs were isolated by successive centrifugation at 4°C: 15 min at 300 g, 10 min at 2000 g, 30 min at 10,000 g and 70 min at 100,000 g (using a Beckman XL-70 centrifuge with a SW28 rotor). EVs pellets were washed in PBS, centrifuged again at 100,000 g for 70 min, resuspended in PBS and stored at 4°C. For all functional experiments, EVs were used immediately after isolation or stored overnight at 4°C and injected the next day. For content analysis, EVs were frozen at −80°C. After EV isolation, EVs numbers and size distribution were measured by NTA using a ZetaView (Particle Metrix, Meerbusch, Germany).

For *in vivo* mouse experiments, EVs were isolated the using the iZON qEV2 size exclusion column (Izon science, Cambridge MA) according to the manufacturer's instructions. After rinsing the columns with PBS, 2 ml of concentrated extracellular medium were applied on top of a qEV column (Izon Science) and 6 ml fractions were collected. For organotropism experiments, four EV-rich fractions (F2, F4, F6, and F8) were pooled, then ultracentrifuged for 1 hr at 100,000 ×g, 4℃ with a SW28 rotor in a Beckman XL-70 centrifuge or concentrated using an Amicon Ultra-4 10 kDa centrifugal filter device (Merck Millipore). Pellets were resuspended in 500 µl PBS. For priming experiment, the most EV-rich fraction was used (F4).

For fluorescent labeling, isolated EVs were incubated with MemBright-Cy3 or Cy5 (*Collot et al., 2018*) at 200 nM (zebrafish) and 500 nM (mice) (final concentration) in PBS for 30 min at room temperature in the dark. Labeled EVs were then rinsed in 15 ml of PBS, centrifuged at 100,000 g with a SW28 rotor in a Beckman XL-70 centrifuge and pellets were resuspended in 50 µl PBS. EVs were used immediately after isolation or stored for a maximum of one night at 4℃ before use.

## Mass spectrometry-based proteomics experiments

Sample preparation of EVs Proteins. A total of 20 mg samples were denatured at 95℃ for 5 min in Laemmli buffer and concentrated in one stacking band using a 5% SDS-PAGE gel. The gel was fixed with 50% ethanol/3% phosphoric acid and stained with colloidal Coomassie Brilliant Blue. The gel bands were cut, washed with ammonium hydrogen carbonate and acetonitrile, reduced and alkylated before trypsin digestion (Promega). The generated peptides were extracted with 60% acetonitrile in 0.1% formic acid followed by a second extraction with 100% acetonitrile. Acetonitrile was evaporated under vacuum and the peptides were resuspended in 10 µl of H20% and 0.1% formic acid before nanoLC-MS/MS analysis.

NanoLC-MS/MS analysis. NanoLC-MS/MS analyses were performed on a nanoACQUITY Ultra-Performance LC system (Waters, Milford, MA) coupled to a Q-Exactive Plus Orbitrap mass spectrometer (ThermoFisher Scientific) equipped with a nanoelectrospray ion source. The solvent system consisted of 0.1% formic acid in water (solvent A) and 0.1% formic acid in acetonitrile (solvent B). Samples were loaded into a Symmetry C18 precolumn (0.18 × 20 mm, 5 µm particle size; Waters) over 3 min in 1% solvent B at a flow rate of 5 µl/min followed by reverse-phase separation (ACQUITY UPLC BEH130 C18, 200 mm x 75 µm id, 1.7 µm particle size; Waters) using a linear gradient ranging from 1% to 35% of solvent B at a flow rate of 450 nl/min. The mass spectrometer was operated in data-dependent acquisition mode by automatically switching between full MS and consecutive MS/MS acquisitions. Survey full scan MS spectra (mass range 300–1800) were acquired in the Orbitrap at a resolution of 70K at 200 m/z with an automatic gain control (AGC) fixed at $3.10^6$ and a maximal injection time set to 50 ms. The 10 most intense peptide ions in each survey scan with a charge state ≥2 were selected for fragmentation. MS/MS spectra were acquired at a resolution of 17.5K at 200 m/z, with a fixed first mass at 100 m/z, AGC was set to $1.10^5$, and the maximal injection time was set to 100 ms. Peptides were fragmented by higher energy collisional dissociation with a normalized collision energy set to 27. Peaks selected for fragmentation were automatically included in a dynamic exclusion list for 60 s. All samples were injected using a randomized and blocked injection sequence (one biological replicate of each group plus pool in each block). To minimize carry-over, a solvent blank injection was performed after each sample. EVs mass spectrometry was performed in triplicate.

Data interpretation. Raw MS data processing was performed using MaxQuant software1 v1.6.7.0 (*Cox et al., 2014*). Peak lists were searched against a database including *Mus musculus* protein sequences extracted from SwissProt (09-10-2019; 17 007 sequences, Taxonomy ID = 10 090). MaxQuant parameters were set as follows: MS tolerance set to 20 ppm for the first search and five ppm for the main search, MS/MS tolerance set to 40 ppm, maximum number of missed cleavages set to 1, Carbamidomethyl (C) set as a fixed modification, Oxidation (M) and Acetyl (Protein N-term) set as variable modifications. False discovery rates (FDR) were estimated based on the number of hits after searching a reverse database and were set to 1% for both peptide spectrum matches (with a minimum length of seven amino acids) and proteins. All other MaxQuant parameters were set as default. Protein intensities were used for label-free quantification. The imputation of the missing values (DetQuantile imputation) and differential data analysis were performed using the open-source ProStaR software (*Wieczorek et al., 2017*). A Limma moderated t-test was applied on the dataset to perform

differential analysis. The adaptive Benjamini-Hochberg procedure was applied to adjust the p-values and FDR values under 1% were achieved.

Complete dataset has been deposited to the ProteomeXchange Consortium via the PRIDE partner repository5 with the dataset identifier PXD020180 (*Deutsch et al., 2020*).

## RNA sequencing

EV pellets were treated with proteinase K (0.05 µg/µl) for 10 min at 37°C. Roche Cocktail Inhibitor was then added to the sample for 10 min at room temperature followed by incubation at 85°C for 5 min. Samples were then incubated with RNase A (0.5 µg/µl) for 20 min at 37°C to degrade unprotected RNA. Total RNAs of isolated EVs was extracted using TRI Reagent (Molecular Research Center). Total RNA Sequencing libraries were prepared with SMARTer Stranded Total RNA-Seq Kit v2 - Pico Input Mammalian (TaKaRa) according to the manufacturer's instructions. Libraries were pooled and sequenced (paired-end 2*75 bp) on a NextSeq500 using the NextSeq 500/550 High Output Kit v2 according to the manufacturer's instructions (Illumina, San Diego, CA, USA). Raw sequencing data generated by the Illumina NextSeq500 instrument were mapped to the mouse reference genome using the hisat2 software (*Kim et al., 2015*). For every sample, quality control was carried out and assessed with the NGS Core Tools FastQC (http://www.bioinformatics.babraham.ac.uk/projects/fastqc/). Read counts were generated with the htseq-count tool of the Python package HTSeq (*Anders et al., 2015*). Differential analysis was performed by the DESEQ2 (*Love et al., 2014*) package of the Bioconductor framework. Detection of significantly up- and down-regulated genes between pairs of conditions based on their log2FC and functional enrichment analyses were performed using STRING v11 (*Szklarczyk et al., 2019*). EVs RNA sequencing was performed in triplicate.

## Lipidomics

EVs were extracted with 2 ml of chloroform/methanol 2/1 v/v and 1 ml water, sonicated for 30 s, vortexed, and centrifuged. Lower organic phase was transferred to a new tube, the upper aqueous phase was re-extracted with 2 ml chloroform. Organic phases were combined and evaporated to dry. Lipid extracts were resuspended in 50 µl of eluent A. Synthetics internals lipid standards (PA 14:1/17:0, PC 17:0/14:1 and PS 17:0/17:0) from Avanti Polar Lipids was added. LC-MS/MS (MRM mode) analyses were performed with a MS model QTRAP 6500 (ABSciex) coupled to an LC system (1290 Infinity II, Agilent). Analyses were achieved in the negative (PA) and in positive (PC) mode; nitrogen was used for the curtain gas (set to 20), gas 1 (set to 20) and gas 2 (set to 10). Needle voltage was at $-$ 4500 or 5500 V without needle heating; the declustering potential was adjusted set at $-$ 172 V or + 40 V. The collision gas was also nitrogen; collision energy is set to $-$ 46 or + 47 eV. The dwell time was set to 30 ms. Reversed phase separations were carried out at 50°C on a Luna C8 150 $\times$ 1 mm column, with 100 Å pore size, 5 µm particles (Phenomenex). Eluent A was isopropanol/$CH_3OH$/$H_2O$ (5/1/4) +0.2% formic acid+0.028% $NH_3$ and eluent B was isopropanol+0.2% formic acid+0.028% $NH_3$. The gradient elution program was as follows: 0–5 min, 30–50% B; 5–30 min, 50–80% B; 31–41 min, 95% B; 42–52 min, 30% B. The flow rate was set at 40 µl/min; 15 µl sample volumes were injected. The areas of LC peaks were determined using MultiQuant software (v3.0, ABSciex) for PA and PC quantification. EVs lipid analysis was performed in triplicate.

## Animal experiments

All animals were housed and handled according to the guidelines of INSERM and the ethical committee of Alsace, France (CREMEAS) (Directive 2010/63/EU on the protection of animals used for scientific purposes). Animal facility agreement number: #C67-482-33. Experimental license for mice: Apafis #4707–20 l6032416407780; experimental license for zebrafish: Apafis #16862–2018121914292754.

### Mouse experiments

Six- to 8-week-old female BalB/c mice (Charles River) were used in all experiments.

Orthotopic breast tumor experiments: Syngenic BalB/c mice were injected in the left fourth mammary gland with 250,000 4T1 mammary tumor cells stably expressing either scramble control shRNA, RalA shRNA, or RalB shRNA and diluted in 50 µl PBS. When tumors became palpable, tumor volume

was assessed by caliper measurements using the formula (width$^2$ × length)/2 (mm$^3$) twice a week for 41 days. At the endpoint of the experiment, tumors and lungs were harvested, weighted, and fixed in formaldehyde. Alternatively, organs were embedded in OCT and frozen at −80°C. In this case, lungs were inflated with OCT before dissection.

### Priming experiments

Mice were injected retro-orbitally with 1.5 × 10$^8$ EVs isolated from 4T1-shControl, shRalA and shRalB cells. Two injections of EVs were performed 2 days apart. PBS was used as a negative control. Subsequently, 4T1-luciferase cells (90.000) were injected via tail vein one day after EV pre-conditioning. After cells injection, the extent of lung metastasis was measured every 3 days for 12 days using non-invasive imaging with IVIS Lumina III (Perkin Elmer). In brief, a D-luciferin solution (purchased from Perkin Elmer and used at 150 mg/kg, according to manufacturer's instructions) was injected intraperitoneally to the isofluorane (Zoetis) anesthetized mice. 5 min after luciferin injection, a bioluminescence image was acquired with an IVIS Lumina III (Perkin Elmer) imaging system and then analyzed using the Living Image software (Perkin Elmer). The rate of total light emission of the lung metastatic area was calculated and expressed as radiance photons counted during the whole acquisition time (5 min) and normalized to the initial radiance photon (photon/second/cm$^2$/sr) measured immediately after 4T1- luciferase cells injection for each mouse (t0).

### EV biodistribution

Mice were injected via retro-orbital venous sinus with 1–4 × 10$^8$ MenBright-Cy3-labeled EVs freshly isolated from 4T1-shControl, shRalA, and shRalB cells. PBS was used as a negative control. Mice were sacrificed 1 hr post-injection to quantify the fluorescence intensity of the organs ex vivo with IVIS Lumina III (Perkin Elmer). Average of fluorescent photons per lung were quantify as radiant efficiency [photon/second/cm$^2$/sr] / [μW/cm$^2$]. For experiment testing the role of CD146 in EV biodistribution, isolated EVs were incubated with CD146 blocking antibody (EPR3208; Abcam; 12 μg/ml) for 30 min at room temperature before injection. For metastasis priming experiments, CD146 was blocked similarly and a rabbit IgG isotype was used as control (Abcam) at an equivalent concentration.

### Zebrafish experiments

At 48 hr post-fertilization (hpf), Tg(Fli1 :GFP) zebrafish embryos were dechorionated and mounted in 0.8% low melting point agarose pad containing 650 mM of tricaine (ethyl-3-aminobenzoate-methanesulfonate). Embryos were injected in the duct of Cuvier with 27.6 nl of Membright Cy5-labeled EVs (at 10$^{10}$ EVs/ml) freshly isolated from 4T1-shControl, shRalA, and shRalB cells with a Nanoject microinjector 2 (Drummond) under a M205 FA stereomicroscope (Leica), using microforged glass capillaries (25–30 mm inner diameter) filled with mineral oil (Sigma). Embryos were imaged with confocal right after injection. For experiment testing the role of CD146, 4T1-isolated EVs were incubated with CD146 blocking antibody (12 μg/ml) for 30 min at room temperature before injection.

## Tissue section and staining

Mouse lungs were incubated overnight in 4% PFA, dehydrated in 100% ethanol for 24 hr, embedded in paraffin, cut in 7-μm-thick sections, dewaxed and rehydrated with 100% Toluene (two washes of 15 min) then incubated in 100–70% alcohol solutions (10 min each) followed by final staining with hematoxylin (Surgipath) for 5 min and washing with tap water. Sections were further processed with differentiation solution (1% HCl in absolute ethanol, for 7 s), followed by washing under tap water for 10 min. Sections were then incubated in eosin (Harris) for 10 s, rinsed and dehydrated in 70–100% alcohol baths with rapid dips in each bath before a final wash in toluene for 15 min and embedded in Eukitt solution (Sigma). Two random distanced sections taken in each of the five lung were analyzed for each mouse. Stitching imaging was performed using an AxioImager (Zeiss) with a ×10 objective. Metastatic surfaces and whole lung surfaces were measured using the Fiji software.

## Caspase 3/7 assay

Mouse tumor samples stored at −80°C are disrupted in a buffer containing Tris HCl pH 7.5, 50 mM, NaCl 150 mM, NP40 1% + Protease Inhibitors cocktail (Complete from Roche) in the presence of 4

zirconium beads, using the Precellis system (Bertin instruments) with two pulses (10'') at 5000 rpm. Protein concentration was measured using Bradford kit (BioRad) and 5 µg was analyzed using the Caspase 3/7 glo kit (Promega) according to manufacturer's instructions. Photons production generated by the luciferase was measured using a luminometer (Berthold Tris Star 2).

## Immunofluorescence

For immunofluorescence on cultured cells, cells were fixed with 4% PFA for 15 min, permeabilized in PBS-Triton 0.1% (Sigma) for 10 min and incubated in 5% normal goat serum for 1 hr. The following primary antibodies were used: ZO-1 (Rabbit, 61–7300; Thermo Fisher Scientific), VE-Cadherin (mouse, 348502; BioLegend), CD63 (rat, D623-3; MBL), RalA (mouse, 610221; BD), RalB (mouse, 04037; Millipore), CD146 (Mouse, P1H12, Thermofisher). The following secondary antibodies were used: goat anti-mouse/rat/rabbit coupled with Alexa Fluor 488, Alexa 555, or Alexa 647 (Invitrogen). Cells were mounted with DAPI-containing Vectashield (Vector Laboratories).

For immunofluorescence on tissue sections, tissues were cut in 7-µm-thick sections, dewaxed for paraffin-embedded tissues and air-dried for frozen tissues. Sections were incubated first in 5% normal goat serum for 2 hr in a humidified container. The following antibodies were used: CD31 (Mouse, 37–0700; Thermo Fisher Scientific), S100A4 A gift from Nona Ambartsumian (Institut for Cancer Biology, Copenhagen, DK-2100, Denmark.), F4/80 (Rat, ab6640; abcam), rabbit monoclonal antibody against Ki67 (Rabbit, RM-9106-S0; Thermo Fisher Scientific), and caspase-3 (Mouse, 966S1; Cell Signaling Technology). Secondary antibodies were similar to the ones used with cells. Nuclei were stained with DAPI (Sigma).

## Imaging and analysis

Imaging on fixed samples. Tissue and cell sections were imaged with a Zeiss Imager Z2 with a $\times 40$ objective (N.A. 1.4) or with an SP5 confocal (Leica) with a $\times 63$ objective (N.A. 1.25). Image analysis and processing were performed using the Fiji software. For endothelial adherent and tight junction analysis, 10 random junctions were analyzed per image (five images per sample) measuring junction width. For Ki67 and Caspase3 imaging, 15 random fields of view were quantified per sample. For EVs imaging, 40–60 random fields of view were imaged on three to four sections per mouse.

Live-cell imaging. For live-cell imaging, cells were seeded on 3.5 cm diameter glass-bottom dishes (MatTek Corporation, Ashland, MA) pre-coated with fibronectin (10 µg/ml; Sigma). Nuclei were labeled with NucBlue Live Ready Probe (Life Technologies, Grand Island, NY). In some experiments, cells were incubated with Lysotracker Deep Red (Thermo Fisher Scientific) at 1 µM for 30 min before imaging. Cells were imaged by confocal microscopy (SP5, Leica) equipped with a thermostated chamber at 37°C with 5% $CO_2$. Image analysis and processing were performed using the Fiji software.

HUVEC cells were seeded in fibronectin (10 µg/ml; Sigma) pre-coated glass bottom culture chambers (LabTek I, Dutscher 055082). Confluent cells were incubated with $2 \times 10^8$ MemBright-labeled EVs in ECGM EV-free medium for 1 hr. Nucleus were labeled using NucBlue (Life Technologies, Grand Island, NY). Cells were imaged by confocal microscopy (SP5 Leica) in a thermostated chamber at 37°C with 5% $CO_2$.

Zebrafish imaging: Confocal imaging was performed on the caudal plexus of zebrafish embryos right after injection with an inverted TCS SP5 with HC PL APO 20X/0,7 IMM Corr CS objective (Leica). Image analysis and processing were performed using the Fiji software.

## Human samples

Human databases: Kaplan-Meier survival curves and statistical analysis of overall survival and gene expression was assessed on the TCGA breast invasive carcinoma cohort (1097 patients) using data generated by the TCGA Research Network: https://www.cancer.gov/tcga.

### Immunohistochemistry

Paraffin sections of 4 µm from metastasic and non-metastasic breast tumours were obtained from CRB-Tumorothèque of the Institut de Cancérologie de l'Ouest (ICO, Saint-Herblain, France) (*Heymann et al., 2020*). Immunohistochemsitry was performed using RalA (BD Transduction #610222, 1/100) and RalB (Sigma WH0005899, 1/400) antibodies on MicroPICell facility (Nantes,

France) Citrate buffer pH6 was used for antigen retrieval 20 min at 96°C (Target Retrieval solution low pH, Dako) and DAB and Hematoxylin staining were revealed using ImPath detection kit (DAB OB Sensitive Detection Kit, ImPath). Whole slides were scanned on Hamamatsu scanner using Nanozoomer Digital Pathology software. Automated computer quantification of DAB staining in perinuclear zones (brown intensity measurement) after automatic nuclei detection with hematoxylin staining in the whole biopsies was performed using Qupath open source software for digital pathology image analysis (*Bankhead et al., 2017*) on MicroPICell platform (Nantes, France). Quantification was further confirmed by manual blinded arbitrary scoring of DAB brown intensity in tumoral zones was performed using a score of 1 for low staining to score of 3 for intense staining.

## Statistical analyses

All results were confirmed in at least two independent experiments. Statistical significance of results was analyzed using the GraphPad Prism program version 5.04. The Shapiro-Wilk normality test was used to confirm the normality of the data. The statistical difference of Gaussian data sets was analyzed using the Student unpaired two-tailed t test, with Welch's correction in case of unequal variances and the one-way ANOVA test followed by a Bonferonni multiple comparison post-test was used for multiple data comparison. For data not following a Gaussian distribution, the Mann-Whitney test was used, and the Kruskal-Wallis test followed by Dunn's Multiple Comparison post-test was used for multiple data comparison. Two Way Anova was used to compare more than one parameters followed by Bonferonni post-test. For analyzing data containing only three measurements, One-Way Anova permutation test followed pairwise permutation test with false detection rate (fdr) correction, using R software (version 3.6.2) was used. Illustrations of these statistical analyses are displayed as the mean +/- standard deviation (SD). p-Values smaller than 0.05 were considered as significant. *, $p < 0.05$, **, $p < 0.01$, ***, $p < 0.001$, ****, $p < 0.0001$.

## Acknowledgements

We thank all members of the Goetz Lab for helpful discussions, in particular Florent Colin for careful reading, as well as Gregory Khelifi and Camille Hergott for animal care. We thank the CRB-Tumorothèque of the Institut de Cancérologie de l'Ouest (ICO, Saint-Herblain, France) and the Cellular and Tissular Imaging Core Facility of Nantes University MicroPICell (SFR-Santé, Nantes, France). We thank Mayeul Collot and Andrey Klymchenko (UMR 7021) for providing the MemBright dye. We thank Monique Dontenwill (CNRS 7213, Strasbourg, France) and Hector Peinado (CNIO, Madrid Spain) for help and discussions. This work was supported by a fellowship from IDEX (University of Strasbourg) and ARC (Association pour le Recherche sur le Cancer) to SG; by grants from La Ligue contre le Cancer, Canceropole Grand-Est, INCa (PLBIO19-291), Plan Cancer (Nanotumor) and Roche to JGG; and by institutional funds from University of Strasbourg and INSERM to JGG, and ANR (to CC, French Proteomics Infrastructure ProFI, ANR-10-INBS-08–03; to NV, ANR-19-CE44-0019). The Metabolome Bordeaux facility was supported by the grant MetaboHUB-ANR-11-INBS-0010. PT and KM were supported by Suttons, Sydney Catalyst, NHMRC, Cancer Council NSW, Cancer Institute NSW and by an Avner Pancreatic Cancer Foundation and Len Ainsworth Pancreatic Research Funds. BM and VM are supported by fellowships from the French Ministry of Science (MESRI).

## Additional information

### Funding

| Funder | Grant reference number | Author |
|---|---|---|
| Institut National Du Cancer | PLBIO19-291 | Jacky G Goetz |
| Ligue Contre le Cancer | | Jacky G Goetz |
| Canceropôle Grand Est | Exosomics | Jacky G Goetz |
| Plan Cancer | Vesmatic | Jacky G Goetz |
| Roche | | Jacky G Goetz |
| Agence Nationale de la Re- | ANR-10-INBS-08-03 | Christine Carapito |

| | | |
|---|---|---|
| cherche | | |
| Agence Nationale de la Re-cherche | ANR-19-CE44-0019 | Nicolas Vitale |
| Agence Nationale de la Re-cherche | ANR-11-INBS- 0010 | Laetitia Fouillen |
| Association pour la Recherche sur le Cancer | | Shima Ghoroghi |
| French ministry of Science | | Benjamin Mary Vincent Mittelheisser |
| Suttons | | Kendelle Murphy Paul Timpson |
| Sydney catalyst | | Kendelle Murphy Paul Timpson |
| NHMRC | | Kendelle Murphy Paul Timpson |
| Cancer Council NSW | | Kendelle Murphy Paul Timpson |
| Pancreatic Cancer Action | | Kendelle Murphy Paul Timpson |
| Pancreatic cancer resource | | Kendelle Murphy Paul Timpson |
| University of Strasbourg | | Jacky G Goetz |
| Region Est | Exosomics | Jacky G Goetz |
| Inserm | | Jacky G Goetz |

The funders had no role in study design, data collection and interpretation, or the decision to submit the work for publication.

## Author contributions

Shima Ghoroghi, Benjamin Mary, Formal analysis, Investigation, Methodology; Annabel Larnicol, Investigation, Methodology; Nandini Asokan, Annick Klein, Naël Osmani, Ignacio Busnelli, Anne-Marie Haeberle, Cathy Royer, Coralie Spiegelhalter, Vincent Mittelheisser, Investigation; François Delalande, Nicodème Paul, Frédéric Gros, Laetitia Fouillen, Gwennan André-Grégoire, Kendelle Murphy, Formal analysis, Investigation; Sébastien Halary, Data curation, Formal analysis; Alexandre Detappe, Paul Timpson, Supervision; Raphaël Carapito, Christine Carapito, Nicolas Vitale, Formal analysis, Supervision; Marcel Blot-Chabaud, Resources; Julie Gavard, Resources, Formal analysis, Supervision; Olivier Lefebvre, Data curation, Formal analysis, Supervision, Investigation; Jacky G Goetz, Conceptualization, Formal analysis, Supervision, Funding acquisition, Investigation, Writing - original draft, Project administration, Writing - review and editing; Vincent Hyenne, Conceptualization, Data curation, Formal analysis, Supervision, Funding acquisition, Investigation, Methodology, Writing - original draft, Writing - review and editing

## Author ORCIDs

Shima Ghoroghi (iD) https://orcid.org/0000-0002-5715-4559
Benjamin Mary (iD) https://orcid.org/0000-0003-0815-842X
Nandini Asokan (iD) https://orcid.org/0000-0003-0505-6487
Naël Osmani (iD) https://orcid.org/0000-0003-1195-753X
François Delalande (iD) https://orcid.org/0000-0003-3590-9874
Nicodème Paul (iD) https://orcid.org/0000-0003-4680-3012
Frédéric Gros (iD) https://orcid.org/0000-0002-6252-4323
Laetitia Fouillen (iD) http://orcid.org/0000-0002-1204-9296
Coralie Spiegelhalter (iD) https://orcid.org/0000-0001-6372-1647
Alexandre Detappe (iD) https://orcid.org/0000-0001-9364-1621
Julie Gavard (iD) https://orcid.org/0000-0002-7985-9007

Christine Carapito [ID] https://orcid.org/0000-0002-0079-319X
Olivier Lefebvre [ID] https://orcid.org/0000-0001-9130-9174
Jacky G Goetz [ID] https://orcid.org/0000-0003-2842-8116
Vincent Hyenne [ID] https://orcid.org/0000-0002-1254-2814

## Ethics

Human subjects: Paraffin sections of 4 µm from metastasic and non-metastasic breast tumours were obtained from CRB-Tumorothèque of the Institut de Cancérologie de l'Ouest (ICO, Saint-Herblain, France) (Heymann et al., 2020). Patients were diagnosed and treated at ICO (Integrated Center for Oncology, St Herblain) for metastatic and non-metastatic breast cancers. Samples were processed and included in the CRB-Tumorothèque ICO upon donor agreement and informed consent. Samples and related information are destroyed at the request of the donor. The CRB-Tumorothèque ICO have been declared to and authorized by the French Research Ministry (Declaration Number: DC-2018-3321). This declaration includes approval by a research ethics committee (CPP "Comité de protection des personnes"), in accordance with the French legislation of the Public Health Code.

Animal experimentation: All animals were housed and handled according to the guidelines of INSERM and the ethical committee of Alsace, France (CREMEAS) (Directive 2010/63/EU on the protection of animals used for scientific purposes). Animal facility agreement number: C67-482-33. Experimental license for mice: Apafis 4707-20l6032416407780; experimental license for zebrafish: Apafis 16862-2018121914292754.

## Decision letter and Author response

Decision letter https://doi.org/10.7554/eLife.61539.sa1
Author response https://doi.org/10.7554/eLife.61539.sa2

# Additional files

## Supplementary files

• Supplementary file 1. EVs RNA analysis. Sheet a: RNAs overexpressed in EVs from 4T1 shCtl cells Vs EVs from shRalA cells Sheet b: RNAs overexpressed in EVs from 4T1 shRalA cells Vs EVs from shCtl cells Sheet c: RNAs overexpressed in EVs from 4T1 shCtl cells Vs EVs from shRalB cells Sheet d: RNAs overexpressed in EVs from 4T1 shRalB cells Vs EVs from shCtl cells

• Supplementary file 2. EVs proteomic analysis. Sheet a: Proteins identified in EVs from 4T1 shCtl cells Sheet b: Proteins overexpressed in EVs from 4T1 shCtl cells Vs EVs from shRalA cells Sheet c: Proteins overexpressed in EVs from 4T1 shRalA cells Vs EVs from shCtl cells Sheet d: Proteins overexpressed in EVs from 4T1 shCtl cells Vs EVs from shRalB cells Sheet e: Proteins overexpressed in EVs from 4T1 shRalB cells Vs EVs from shCtl cells Sheet f: Proteins overexpressed in EVs from 4T1 shCtl cells Vs EVs from shRalA cells and EVs from shRalB cells

## Data availability

Sequencing and mass spectrometry data have been deposited in the EV-Track database.

The following previously published dataset was used:

| Author(s) | Year | Dataset title | Dataset URL | Database and Identifier |
|---|---|---|---|---|
| The Cancer Genome Atlas Network | 2012 | Comprehensive molecular portraits of human breast tumours | https://www.cancer.gov/tcga | TCGA breast invasive carcinoma cohort (10 97 patients), 10.1038/ nature11412 |

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
