## [Decision Letter]

**Acceptance summary:**

Overall, we find that this a very complete and thorough study of the role of 2 small GTPases, RalA and RalB, in extracellular vesicle (EV) release and breast carcinoma progression. We believe the main advance of this manuscript is to describe a signaling role for the GTPases RalA and RalB in regulating phospholipase D (PLD) and multivesicular body function to regulate exosome biogenesis and organ-targeting of exosomes, with specific relevance to lung cancer metastasis. Notably, the work has clinical relevance, given the finding that RalA, RalB and CD146 levels are indicators poor prognosis for breast cancer patients. Overall these are interesting observations that should impact multiple fields of study.

**Decision letter after peer review:**

Thank you for submitting your article "Ral GTPases promote metastasis by controlling biogenesis and organ colonization of exosomes" for consideration by *eLife*. Your article has been reviewed by three peer reviewers, one of whom is a member of our Board of Reviewing Editors, and the evaluation has been overseen by Suzanne Pfeffer as the Senior Editor. The following individual involved in review of your submission has agreed to reveal their identity: Aaron Hobbs (Reviewer #3).

The reviewers have discussed the reviews with one another and the Reviewing Editor has drafted this decision to help you prepare a revised submission.

Summary:

The role of extracellular vesicles (EVs) such as exosomes as factors potentiating metastasis by solid tumors has attracted considerable recent interest. The article by Ghoroghi et al. is a very complete and thorough study of the role of 2 small GTPases, RalA and RalB, in extracellular vesicle (EV) release and breast carcinoma progression. The main advance of this manuscript is to describe a signaling role for the GTPases RalA and RalB in regulating phospholipase D (PLD) and multivesicular body function to regulate exosome biogenesis. They also show that RalA and RalB regulate expression of MCAM/CD146 on EVs, and that reduced levels of CD146 on EVs affects efficiency of lung cancer metastasis. Finally, they show RalA, RalB and CD146 levels are indicators poor prognosis for breast cancer patients. Overall these are interesting observations with clinical relevance. The study is extremely carefully performed, in general, with appropriate controls and conclusions. There are a limited number of weaknesses that need to be addressed.

Essential revisions:

1) Figure 3C shows that RALA and RALB have functions where they do not always act in series with each other. They may for exosome excretion, but not necessarily for proliferation. For Figure 3—figure supplement 1C, this experiment needs to be longer than 72 hours for a proliferation assay to be truly convincing. While interesting that proliferation results differ between mice and growth on plastic, a 10 day growth experiment would lead to greater conviction that this is a sustained difference. Figure 3D better agrees with Figure 3—figure supplement 1C, where loss of RALB gives greater proliferation over control, albeit not as great as RALA loss. Was Figure 3D performed at 12 days? The variable time points make it difficult to assess the role of RALA/B on growth.

2) Figure 4 carefully describes why RALA/B deficient EVs fail to prime regions for metastasis. As the EVs are not directed to those locations, the contents fail to increase permeability of the regions. However, statistically significant does not mean biologically meaningful; specifically, looking at Figure 4F and 4G, knockdown of RALA/B had little effect on EV internalization. One question that is not answered is whether RALA/B are acting in series or parallel. Concurrent depletion of both RALA and RALB in the types of experiment performed in Figure 4 would help answer this question. If depletion of both isoforms is greater than either alone, then it implies they act in parallel.

3) How are RALA/B activated in the context of 4T1 cells? The use of the RAL inhibitor in Figure 1B is most effective in cell lines without RAS mutations. Panc1 and MD231 cells have KRAS mutations, yet RALGEF inhibitors are least effective. At the concentrations used, all RALGEF activity should be inhibited. Is RAL expression levels similar between cell lines?

4) The logic for targeting PLD1 seems rushed. There are several RAL effectors, and while the data on PLD1 are compelling, it is unclear why only this effector was selected. Were other effectors tested? The Hyenne paper, which is cited throughout, provided stronger logic for connecting these pathways. The statement in the Discussion, "Our work further identifies PLD as the most likely effector acting downstream of RAL to control exosome section" is incorrect as stated. This work identified PLD1 as A downstream effector important for exosome secretion. To state that it is the most likely effector is an overstatement as no other RAL effectors were tested.

5) The use of either PLD inhibitor at 10 μm will have significant inhibition of both PLD1 and PLD2 regardless which inhibitor is being tested, as well as additional off-target effects. Given both inhibitors have nanomolar IC50 values for their specific PLD isoform and low μM IC50 values for the related isoform, these experiments are inconclusive as presented. This section of the text should be reworked to account for the lack of isoform selectivity at the concentrations of inhibitor used.

a) siRNA or shRNA could be used to validate the PLD1/2 inhibitor data.

6) In all cases when using inhibitors, there are no western blots or additional validation showing target inhibition in any cell line. In this case, the use of such high concentrations of inhibitor likely resulted in inhibition of the target proteins. However, such high concentrations often result in off-target effects. Determining the lowest efficacious dose to inhibit target function should have been performed and used throughout the experiments.

7) Would loss of MCAM expression, via direct knockdown, result in EVs that are unable to permeabilize HUVEC cells, as in Figure 4A/B? Alternatively, does loss of MCAM expression (or blocking via anti-CD146) result in decreased metastasis, similar to RALA/B knockdown? While Figure 5G shows a decrease in EV localization to lungs on treatment with Anti-CD146, the rate of metastatic lesion formation was not assessed when CD146 was blocked.

8) In a previous study (Hyenne et al., 2015) the authors already demonstrated a role for RalA and RalB in controlling MVB and EV secretion in the 4T1 breast cancer model, in a process evolutionarily conserved through nematodes. They also wrote a follow-up review article describing considerable evidence in the literature for RalA controling PLD and ARF6 function to control EV secretion. This diminishes the novelty and interest of the first part of the study, which could be reduced. The more novel and thought-provoking parts of the study are in the definition of the relationship between the RAL proteins and CD146, and the identification of unique properties of the exosomes produced in cells with manipulated RAL proteins (for instance, in regard to influencing exosome permeability). These could be better emphasized.

9) Almost the entire mechanistic model is based on the work of a single breast cancer cell line, 4T1, which has been described by some as triple negative. This is a significant weakness, as there may be features of that line that make it uncharacteristic of breast cancer in general. At least some of the key conclusions should be functionally confirmed in an additional cell model.

10) In addition, breast cancer cells fall into multiple different subtypes, which have different metastatic propensities and gene expression patterns. Is the functional relationship between the expression of RALA/B and CD146 in exosomes observed in just triple negative cells, or in other subtypes?

11) In Figure 5H, K-M analysis of TCGA shows a weak relationship between MCAM expression and survival. Were the tumors analyzed segregated by tumor subtype? Were data corrected for tumor stage? This is important, as the MCAM staining pattern may reflect a propensity for MCAM expression in tumors at a late stage, or subtypes with a poorer prognosis.

12) The authors provide observations suggesting that RalA and B control primarily the EV secretion pathway involving Multivesicular bodies, hence leading to exosome secretion. This is mainly demonstrated by observation of a decrease in MVBs upon knock-down of RalA/B, demonstrated by thorough electron microscopy analyses. This is correlative, rather than truly demonstrative, but the best one can do so far. In most experiments, the authors use EVs isolated by a relatively crude method, ultracentrifugation, that co-isolates non-specific components, and they do not analyse larger EVs that can be recovered at lower centrifugation speed, thus an effect of RalA/B on these non-exosomal EVs cannot be excluded. The EVs are only characterized by their number (NTA counting), which is not very precise, and not consistent with guidelines of ISEV (MISEV 2018, J Extracell Vesicles 2018, 7: 1535750).

Maybe the authors can argue that they did perform more complete analyses of their 4T1 EVs in a previous article? Did they use the EV-TRACK website to verify their experimental EV isolation and characterization set up? Of note, the authors also perform quantitative proteomic and RNomic analyses, which gives a better characterization of EVs. The protein composition and its change upon RalA/B KD could have been used to try to confirm (or not) the MVB origin of the EVs controlled by RalA or RalB, but it is not crucial for the message.

Alternatively, to demonstrate that the CD146-bearing EVs that carry the prometastatic function are bona fide exosomes, the authors could have shown its localization in the cells, upon or not RalA/B depletion, and show if a drastic change in ratio of localization in MVBs vs the PM occurs, but this would be an additional study, not necessary for the current paper.

13) The authors insist that RalA/B control exosome secretion, and discuss the bases and limitations of their demonstration properly. The summarizing schemes (Figure 2F and Figure 5I) of their model show the release of RalA/B-depent pro-metastatic exosomes, and RalA/B-independent exosomes which are not pro-metastatic. However, EVs that are released in the absence of RalA/B could instead be formed at the plasma membrane, and correspond to ectosomes. Nothing in this study demonstrates the origin of RalA/B-independent EVs, thus PM-derived EVs should be represented in the scheme.

---

## [Author Response]

Essential revisions:1) Figure 3C shows that RALA and RALB have functions where they do not always act in series with each other. They may for exosome excretion, but not necessarily for proliferation. For Figure 3—figure supplement 1C, this experiment needs to be longer than 72 hours for a proliferation assay to be truly convincing. While interesting that proliferation results differ between mice and growth on plastic, a 10 day growth experiment would lead to greater conviction that this is a sustained difference. Figure 3D better agrees with Figure 3—figure supplement 1C, where loss of RALB gives greater proliferation over control, albeit not as great as RALA loss. Was Figure 3D performed at 12 days? The variable time points make it difficult to assess the role of RALA/B on growth.

As the reviewer pointed out, our initial manuscript reports that in vivo tumor growth (Figure 3D; at 41 days) differs form in vitro proliferation assays (Figure 3—figure supplement 1C; over 72h). In details, while loss of RalA consistently shows a trend to increased growth in both conditions, loss of RalB shows a slight increase in cell proliferation in vitro but a reduced tumor growth in vivo. To better understand this discrepancy, we carefully analyzed the cell cycle of shRal cells by flow cytometry. We observed a significant decrease in the proportion of cells in the sub-G1 phase in shRalA cells compared to control, potentially explaining the overall increased proliferation observed in this condition. However, the proportion of cells in each phase of the cycle was similar in shRalB and shCtl cells. Overall, this analysis further illustrates the importance of in vivo experiments, and the difficulty to reconcile in vitro with in vivo situations (where microenvironment and non-cell intrinsic parameters are likely to significantly alter these behaviors and impact tumor growth). We have now included the cell cycle analysis in Figure 3—figure supplement 1D and mentioned it in the manuscript.

2) Figure 4 carefully describes why RALA/B deficient EVs fail to prime regions for metastasis. As the EVs are not directed to those locations, the contents fail to increase permeability of the regions. However, statistically significant does not mean biologically meaningful; specifically, looking at Figure 4F and 4G, knockdown of RALA/B had little effect on EV internalization. One question that is not answered is whether RALA/B are acting in series or parallel. Concurrent depletion of both RALA and RALB in the types of experiment performed in Figure 4 would help answer this question. If depletion of both isoforms is greater than either alone, then it implies they act in parallel.

We would like to split the reviewer’s comment in 2 parts:

1) how do we explain the relatively modest defects in shRal EVs internalization in Figure 4F/G (in vitro-in zebrafish) compared to the (more important) effect on metastasis priming (Figure 4D) ?

2) Do RalA and RalB act in series or in parallel?

Regarding the (i) first question, we would like to insist on two aspects that are likely to be driven by the inherent differences of the experimental approaches used: first, the shRal EVs targeting defects are stronger in mice lungs (Figure 4D/E) than in vitro (Figure 4F) or in zebrafish (ZF, Figure 4E). This read-out (targeting of EVS to the lung parenchyma) is not exactly comparable to the assays used in vitro and in ZF (internalization in an endothelium). Nevertheless, the experimental approach used in such homologous model is very close to the one used for the metastatic niche priming model. Interestingly, in both cases, shRal EVs show comparably high deficiencies (60-70% decrease) which demonstrates that EV targeting/internalization in mice is directly linked to its effect on metastatic niche priming. In contrast, while assessing internalization of EVs in HUVECs or ZF endothelium offers a level of resolution that can’t be reached in mouse models, comparing these different assays of EV internalization also suggests that the mechanisms involved in the internalization of circulating EVs in mice are more complex that the one taking place in heterologous systems. We hypothesize that HUVECs or ZF take up EVs in a less specific manner and are therefore less sensitive to targeting of specific adhesion receptors. Second, Ral depletion can affect adhesive-independent properties of breast tumor EVs able to influence premetastatic niche priming. Indeed, several proteins and multiple RNAs show alteration of their expression levels in shRal EVs (Figure 5A-D). Some of these cargoes could contribute to the formation of a premetastatic niche without affecting the adhesion properties of the EVs.

The (ii) second question regarding the potential redundancy of RalA and RalB has puzzled Ral experts for more than two decades. Depending on the context and biological effect, Ral GTPases can have redundant, parallel or antagonist functions. To address this question and determine whether RalA and RalB act in series or in parallel in EV secretion and function, we first generated 4T1 cells with a double knock-down (KD) for both RalA and RalB . We then determined how their co-depletion affects EV secretion. Co-depletion of both RalA and RalB further decreases EV secretion (when compared to single RalA or RalB depletion). To better understand how they could function, we decided to analyze the phenotypes of MVBs in double KD cells by electron microscopy (based on high pressure freezing). Here again, we observed that co-depleting Ral GTPases further decreases the density of MVBs in the cytoplasm when compared to single KD. Therefore, co-depletion of Ral GTPases worsens individual phenotypes, which implies that RalA and RalB act partially in parallel (or independent) ways. However, the decrease observed does not correspond to the sum of their individual depletion, suggesting that they still act on a similar exosomal secretion pathway.

Understanding precisely how RalA and RalB cooperate in exosome secretion and function would imply a much deeper epistatic analysis. While we have now initiated such analysis, which will certainly require a significant amount of work, we hope the reviewers understand that such analysis goes beyond this study. While we are thankful to the reviewers for raising and confirming this important question, we would like to ask for their understanding as we favor a more thorough analysis and thus exclude these incomplete data from the current study. At this stage, and without an extensive study, it is yet impossible to claim any definitive conclusion, which our future work will allow to solve confidently.

3) How are RALA/B activated in the context of 4T1 cells? The use of the RAL inhibitor in Figure 1B is most effective in cell lines without RAS mutations. Panc1 and MD231 cells have KRAS mutations, yet RALGEF inhibitors are least effective. At the concentrations used, all RALGEF activity should be inhibited. Is RAL expression levels similar between cell lines?

Regarding the first question, we currently ignore how RalA/B are activated in 4T1 cells. However, we speculate that they could be activated by the RAS pathway, which is active in 4T1 cells and required for 4T1 cells aggressivity. Indeed, 4T1 cells express KRAS and HRAS (Xi et al., 2017; Hu et al., Nuc. Acid Res. 2019) and the RAS pathway is active in 4T1 cells, as assessed by BRAF and erk phosphorylation status (Phan et al., 2013). Importantly, both 4T1 migration abilities and tumor progression depend on KRAS expression levels (Xi et al., 2017; Booth et al., 2017).

To address the reviewer’s last question, we measured RAL expression levels in several cell lines by western blot (Author response image 1). Interestingly, the 4T1 cell lines thoroughly used in our study displays the highest Ral expression levels. In addition, as pointed out by the reviewer, a higher expression of RalA could explain that the effect of Ral inhibitors on EV secretion is higher in 4T1 cells than in Panc1 or MDAMD231 cells.

**Author response image 1. sa2fig1:** RAL expression. RalA and RalB expression assessed by western blots (Left in multiple cell types and quantified (Right). Each dot represents one experiment.

4) The logic for targeting PLD1 seems rushed. There are several RAL effectors, and while the data on PLD1 are compelling, it is unclear why only this effector was selected. Were other effectors tested? The Hyenne paper, which is cited throughout, provided stronger logic for connecting these pathways. The statement in the Discussion, "Our work further identifies PLD as the most likely effector acting downstream of RAL to control exosome section" is incorrect as stated. This work identified PLD1 as A downstream effector important for exosome secretion. To state that it is the most likely effector is an overstatement as no other RAL effectors were tested.

When dissecting the mechanistic effects of RalA/B on exosome secretion, we have indeed used a strategy that is based on published evidence for targeting PLDs as Ral effectors, in the context of exosome secretion. Thus, while this reviewer is partly correct since we have only tested PLDs and not other Ral effectors, such decision had been carefully thought based on arguments that are described in the manuscript and summarized here.

Among all known Ral effectors, PLDs had been uniquely connected to exosome secretion, through the work of Pascal Zimmermann’s group (Ghossoub et al., 2014), and were therefore the most obvious candidates to be targeted. Interestingly, recent independent studies from Borel and colleagues confirmed that PLDs affect EV secretion levels in a Phosphatidic Acid dependent manner in prostate cancer cells (Borel et al., BBA 2020). While other Ral effectors (such as RalBP1, Filamin or exocyst) have, to our knowledge, never been shown to alter EV secretion, they are indeed potential additional candidates whose targeting would require additional work that would go beyond this study.

Taking this into account, we have now modified the way we introduce PLD1 and toned down our conclusions.

5) The use of either PLD inhibitor at 10 μm will have significant inhibition of both PLD1 and PLD2 regardless which inhibitor is being tested, as well as additional off-target effects. Given both inhibitors have nanomolar IC50 values for their specific PLD isoform and low μM IC50 values for the related isoform, these experiments are inconclusive as presented. This section of the text should be reworked to account for the lack of isoform selectivity at the concentrations of inhibitor used.a) siRNA or shRNA could be used to validate the PLD1/2 inhibitor data.

We have now added a sentence in the Discussion to emphasize the potential lack of isoform selectivity. Nevertheless, we would like to emphasize that the concentration that have been used had been published before (Figure 6 of Ghossoub et al., 2014) when demonstrating that PLD2 is specifically involved in ARF6 exosome biogenesis.

Furthermore, as suggested by the reviewer, we undertook experiments with siRNA against PLD1/2. Unfortunately, these experiments failed as we did not manage to specifically detect PLD isoforms by western blotting using commercially available antibodies (data not shown).

Finally, to further demonstrate that we can indeed observe defects related to each PLD isoform, we subjected cells to decreasing concentrations of PLD inhibitors (in order to reach nanomolar IC50 values) and observed, as expected, a dose-dependent effect of the inhibition on EV secretion levels (Author response image 2, see below, #6). These results show that EV secretion levels can be tuned with nanomolar concentrations of PLD inhibitors. Thus, although both PLDs are involved in controlling EV secretion levels (validated here), only PLD1 localization to endosome is perturbed in absence of RalA/B and thus responsible of Ral dependent EV secretion. Nevertheless, in the Discussion, we acknowledge the possibility of off target effects at high inhibitor concentrations.

**Author response image 2. sa2fig2:** Effects of PLD doses on EV secretion. Each experiment was repeated three times on 4T1 cells.

6) In all cases when using inhibitors, there are no western blots or additional validation showing target inhibition in any cell line. In this case, the use of such high concentrations of inhibitor likely resulted in inhibition of the target proteins. However, such high concentrations often result in off-target effects. Determining the lowest efficacious dose to inhibit target function should have been performed and used throughout the experiments.

While we appreciate the fact that high concentrations might lead to off-target effects, we would like to emphasize again that these concentrations (for both Ral and PLD inhibition) have been used and published in the past (Yan et al., 2014; Ghossoub et al., 2014) and demonstrated their efficacy and selectivity. As suggested by the reviewer, we now used different concentrations of PLD inhibitors (10nM, 100nM, 1µM and 10µM) and analyzed their effect on EV secretion by NTA. As shown on Author response image 2, there is a clear dose dependent effect, using both inhibitors. These results show that EV secretion levels can be tuned with nanomolar concentrations of PLD inhibitors. Thus, although both PLDs are involved in controlling EV secretion levels (validated here), only PLD1 localization to endosome is perturbed in absence of RalA/B and thus responsible of Ral-dependent EV secretion.

7) Would loss of MCAM expression, via direct knockdown, result in EVs that are unable to permeabilize HUVEC cells, as in Figure 4A/B? Alternatively, does loss of MCAM expression (or blocking via anti-CD146) result in decreased metastasis, similar to RALA/B knockdown? While Figure 5G shows a decrease in EV localization to lungs on treatment with Anti-CD146, the rate of metastatic lesion formation was not assessed when CD146 was blocked.

We thank the reviewer for raising this point and agree that such aspect is determinant to validate our model. In order to test whether blocking CD146 would tune tumor metastasis, we set out a challenging experimental approach allowing to explore the role of EV-bound CD146 in metastatic niches priming. To do so, we set up a classical “priming” experiment where 4T1 EVs are blocked with the anti-CD146 antibody. As shown in Figure 5H and Figure 4C, pre-injection of 4T1 EVs increases lung metastasis of 4T1 cells compared to PBS treatment. However, when pre-injecting a similar amount of 4T1 EVs treated with the blocking anti-CD146 antibody, we observed a striking reduction in lung metatstatic burden, nearing PBS-treated levels (14 mice per group in two independent experiments). This experiment solidly demonstrates that CD146, exposed at the surface of EVs and whose enrichment is tuned by RalA/B, is a key molecular determinant of EV-mediated metastasis. Together with our previous results, it strongly suggests that CD146 molecules present at the surface of tumor EVs favor lung colonization by EVs, likely by favoring binding to the endothelium, thereby allowing pre-metastatic niche formation. As this result provides a new and solid mechanistic insight into the molecular machineries tuned by RalA/B levels and thus consolidates and improves our earlier observations, we decided to add it in Figure 5I of our manuscript and discuss it in the main text.

8) In a previous study (Hyenne et al., 2015) the authors already demonstrated a role for RalA and RalB in controlling MVB and EV secretion in the 4T1 breast cancer model, in a process evolutionarily conserved through nematodes. They also wrote a follow-up review article describing considerable evidence in the literature for RalA controling PLD and ARF6 function to control EV secretion. This diminishes the novelty and interest of the first part of the study, which could be reduced. The more novel and thought-provoking parts of the study are in the definition of the relationship between the RAL proteins and CD146, and the identification of unique properties of the exosomes produced in cells with manipulated RAL proteins (for instance, in regard to influencing exosome permeability). These could be better emphasized.

We can’t agree more with this reviewer when he claims that the most novel aspect of our study is the relationship between Ral and CD146 levels and role on EVs. In turn, and with the newly added mechanistic data (Figure 5H), we’ve further put the emphasis on this novel aspect of our work.

Moreover, we wish to maintain the second additional exciting part of this work, which is the careful molecular and cell biological dissection of Ral function in exosome secretion. Our previous description of PLD involvement downstream of Ral (Hyenne at al, 15 + 16) was mostly speculative while, we now provide, within this manuscript, experimental evidences of the molecular links between Ral and PLD1. In the light of the new experiments, we have now modified the Abstract accordingly and have incorporated our latest results on CD146 in priming metastasis as Figure 5I (See Comment #7).

9) Almost the entire mechanistic model is based on the work of a single breast cancer cell line, 4T1, which has been described by some as triple negative. This is a significant weakness, as there may be features of that line that make it uncharacteristic of breast cancer in general. At least some of the key conclusions should be functionally confirmed in an additional cell model.

With all due respect, we wish to point out that we also had observed that Ral inhibition leads to a significant decrease in EV secretion levels in pancreatic, melanoma and breast cancer cells (Figure 1B). Thus, together with our previous work performed in *C. elegans* (Hyenne et al., 2015), this suggests that the role of Ral in EV secretion can be extended is likely to be universal or can be at least extended to other models and is conserved in evolution. In addition, additional evidence had been obtained when correlating Ral levels with metastasis in our cohorts (Figure 3A, B) or via access to TCGA public databases (that contain various breast cancer subtypes, see #10), which further demonstrates that high Ral levels (or CD146) correlate with metastasis and poor survival. Thus, altogether, this demonstrates that our discovery can be extended to other breast cancer subtypes. While we appreciate that our in vivo mouse approaches are restricted to the TNBC 4T1 model, such experiments are heavy and challenging and should be seen as an advantage, rather than a pitfall of our work. Further in vivo experiments performed by us (or others) will provide additional insights on the universality of such pathway, but we humbly feel that this beyond the scope of this study. Finally, we now provide additional demonstration that Ral inhibition is effective in other model, including three other breast cancer cell lines (see Point 10; Figure 1—figure supplement 1C).

10) In addition, breast cancer cells fall into multiple different subtypes, which have different metastatic propensities and gene expression patterns. Is the functional relationship between the expression of RALA/B and CD146 in exosomes observed in just triple negative cells, or in other subtypes?

To address this question, we analyzed the role of Ral GTPases in EV secretion by three other breast cancer cell lines with distinct characteristics: MCF7 (human, estrogen receptor^+^, progesterone receptor^+^, HER2^-^), SKBR3 (human, HER2^+^) and D2A1 (murine, estrogen receptor^+^, HER2^+^). In all cases, we found that Ral inhibition with RBC8 drug induces a significant decrease in EV secretion (Figure 1—figure supplement 1C). This further demonstrates that all breast cancer types release Ral dependent EVs. This new data is now added as Figure 1—figure supplement 1C and the associated description added in the manuscript accordingly.

11) In Figure 5H, K-M analysis of TCGA shows a weak relationship between MCAM expression and survival. Were the tumors analyzed segregated by tumor subtype? Were data corrected for tumor stage? This is important, as the MCAM staining pattern may reflect a propensity for MCAM expression in tumors at a late stage, or subtypes with a poorer prognosis.

We followed the reviewer recommendation and re-analyzed TCGA data, segregating the patients by tumor subtypes and by tumor stages (Author response image 3). This analysis indeed revealed differences between tumor subtypes. High CD146 expression tends to correlate with decreased survival probability in three subtypes (basal, Her2+ and Luminal A) and with increased survival in one subtype (Luminal B). However, statistical significance was only reached for the luminal A subtype. Importantly, high CD146 expression correlates with a decreased survival probability in stage III breast cancer, while stages I and II are inconclusive, due to low patients number.

We performed similar analysis for RalA and RalB and found that elevated expression of both GTPases show a tendency to correlate with decreased survival in all breast cancer subtypes and stages (with the notable exceptions of basal subtype where high RalB correlates with enhanced survival and Luminal where no clear correlation is observed for RalB). In conclusion, at this stage, while Ral levels clearly correlate with poor survival, such correlation for MCAM/CD146 will require additional work.

**Author response image 3. sa2fig3:** Segmentation of TCGA data.

12) The authors provide observations suggesting that RalA and B control primarily the EV secretion pathway involving Multivesicular bodies, hence leading to exosome secretion. This is mainly demonstrated by observation of a decrease in MVBs upon knock-down of RalA/B, demonstrated by thorough electron microscopy analyses. This is correlative, rather than truly demonstrative, but the best one can do so far. In most experiments, the authors use EVs isolated by a relatively crude method, ultracentrifugation, that co-isolates non-specific components, and they do not analyse larger EVs that can be recovered at lower centrifugation speed, thus an effect of RalA/B on these non-exosomal EVs cannot be excluded. The EVs are only characterized by their number (NTA counting), which is not very precise, and not consistent with guidelines of ISEV (MISEV 2018, J Extracell Vesicles 2018, 7: 1535750).Maybe the authors can argue that they did perform more complete analyses of their 4T1 EVs in a previous article? Did they use the EV-TRACK website to verify their experimental EV isolation and characterization set up? Of note, the authors also perform quantitative proteomic and RNomic analyses, which gives a better characterization of EVs. The protein composition and its change upon RalA/B KD could have been used to try to confirm (or not) the MVB origin of the EVs controlled by RalA or RalB, but it is not crucial for the message.Alternatively, to demonstrate that the CD146-bearing EVs that carry the prometastatic function are bona fide exosomes, the authors could have shown its localization in the cells, upon or not RalA/B depletion, and show if a drastic change in ratio of localization in MVBs vs the PM occurs, but this would be an additional study, not necessary for the current paper.

As pointed by this reviewer, we indeed partially characterized 4T1 EVs in a previous study (Hyenne et al., 2015). More precisely, we characterized and quantified 4T1 EVs (secreted by control, shRalA and shRalB cells) by electron microscopy and western blots using different markers (TSG101, CD63, Hsc70 and Alix). In the present study, we used a complementary quantification method, Nanoparticle Tracking Analysis, to assess EV size and concentrations, and characterized the content of these EVs by mass spectrometry, RNA sequencing and lipidomics (measuring PC and PA levels). Therefore, we humbly disagree with the reviewer and rather think that we deeply characterized 4T1 EVs secreted by control and Ral depleted cells. Our data were indeed submitted to EV-Track during the revision process. Regarding the MVB origin of EVs secreted in a Ral-dependent manner, we agree with the reviewer that our observations are mostly correlative. We cannot fully exclude that some of these EVs could originate from the plasma membrane and we now acknowledge this in the manuscript.

However, the mass spectrometry analysis revealed a large proportion of endosomal proteins, and this aspect is now further emphasized in the Results section. Following the reviewer’s recommendations, we now carefully assessed the localization of CD146 in 4T1 cells by immunofluorescence. We found that CD146 is present both at the plasma membrane and in CD63+ MVBs/late endosomes. Since we had shown that both RalA and RalB colocalize with CD63 (Figure 1C), this suggests that CD146 present on EVs could originate from MVBs, where its loading would depend on Ral GTPases. However, as pointed by the reviewer, describing such trafficking in details would require an independent study. We have added CD146 endosomal localization in 4T1 cells in Figure 5—figure supplement 2C.

13) The authors insist that RalA/B control exosome secretion, and discuss the bases and limitations of their demonstration properly. The summarizing schemes (Figure 2F and Figure 5I) of their model show the release of RalA/B-depent pro-metastatic exosomes, and RalA/B-independent exosomes which are not pro-metastatic. However, EVs that are released in the absence of RalA/B could instead be formed at the plasma membrane, and correspond to ectosomes. Nothing in this study demonstrates the origin of RalA/B-independent EVs, thus PM-derived EVs should be represented in the scheme.

We agree with the reviewer and modified our scheme in Figure 2F accordingly. However, for the sake of clarity of our final take-home message, we would appreciate if the reviewers agree that we focus on exosome secretion (Ral dependent and Ral independent) in the final model from Figure 5J, and thus exclude Ral-independent microvesicles from this graph (as these have not been tested in our study).